# Virtual substrate method for nanomaterials characterization

Bo Da[1,2,3], Jiangwei Liu[1], Mahito Yamamoto[4], Yoshihiro Ueda[5], Kazuyuki Watanabe[5], Nguyen Thanh Cuong[1,4], Songlin Li[4], Kazuhito Tsukagoshi[4], Hideki Yoshikawa[2], Hideo Iwai[2], Shigeo Tanuma[2], Hongxuan Guo[6], Zhaoshun Gao[7], Xia Sun[8] & Zejun Ding[8]

Characterization techniques available for bulk or thin-film solid-state materials have been extended to substrate-supported nanomaterials, but generally non-quantitatively. This is because the nanomaterial signals are inevitably buried in the signals from the underlying substrate in common reflection-configuration techniques. Here, we propose a virtual substrate method, inspired by the four-point probe technique for resistance measurement as well as the chop-nod method in infrared astronomy, to characterize nanomaterials without the influence of underlying substrate signals from four interrelated measurements. By implementing this method in secondary electron (SE) microscopy, a SE spectrum (white electrons) associated with the reflectivity difference between two different substrates can be tracked and controlled. The SE spectrum is used to quantitatively investigate the covering nanomaterial based on subtle changes in the transmission of the nanomaterial with high efficiency rivalling that of conventional core-level electrons. The virtual substrate method represents a benchmark for surface analysis to provide 'free-standing' information about supported nanomaterials.

[1] International Center for Young Scientists, National Institute for Materials Science, Tsukuba, Ibaraki 305-0047, Japan. [2] Surface Chemical Analysis Group, Nano Characterization Unit, National Institute for Materials Science, Tsukuba, Ibaraki 305-0047, Japan. [3] Magnet Materials Group, Center for Materials Research by Information Integration, National Institute for Materials Science, Tsukuba, Ibaraki 305-0047, Japan. [4] International Center for Materials Nanoarchitectonics, National Institute for Materials Science, Tsukuba, Ibaraki 305-0044, Japan. [5] Department of Physics, Tokyo University of Science, Tokyo 162-8601, Japan. [6] Center for Nanoscale Science and Technology, National Institute of Standards and Technology, Gaithersburg, Maryland 20899, USA. [7] National Institute for Materials Science, Tsukuba, Ibaraki 305-0047, Japan. [8] Department of Physics, University of Science and Technology of China, Hefei, Auhui 230026, China. Correspondence and requests for materials should be addressed to B.D. (email: DA.Bo@nims.go.jp) or to H.Y. (email: YOSHIKAWA.Hideki@nims.go.jp) or to S.T. (email: TANUMA.Shigeo@nims.go.jp).

Nanomaterials are materials at the smallest scale and near the forefront of research in natural sciences. Nanomaterials show great potential to revolutionize industry, medicine and computing, and improve our understanding and conservation of nature. Various types of nanomaterials have been subjected to many chemical and physical analyses typically applied to bulk or film solid-state materials[1–3]. However, most of these analysis tools are unsuitable for substrate-supported nanomaterial samples because of the influence of underlying substrate signals, particularly for techniques using reflection configuration[4]. Even electron-based approaches, represented by surface analysis techniques such as X-ray photoelectron spectroscopy and Auger electron spectroscopy (AES), whose probing depths are at the nanoscale level, have been limited by this problem. Surface analysis techniques, which typically use reflection configuration are powerful tools to quantitatively obtain elemental composition and chemical-state information of materials[5–7] and have been applied to intact substrate-supported nanomaterial samples[8–10]. Not surprisingly, only qualitative information about nanomaterials can be obtained using traditional operating procedures because of the influence from substrate signals. Such a substrate contribution cannot simply be removed by purposely decreasing the probing depth, because doing this causes the obtained information to be related to the properties of the surface atomic layer of the nanomaterial, rather than the overall properties of the entire nanomaterial. Generally, the overall properties of the entire nanomaterial can only be measured by techniques using transmission configuration. Therefore, for techniques using reflection configuration, there is a need for a new method that is able to obtain information about entire nanomaterials without influence from substrate signals even when the nanomaterial is supported by a substrate.

The method that is currently most widely used to extract nanomaterial information from measurements obtained for substrate-supported nanomaterial samples can be summarized as a two-point probe method, in which traditional data-processing techniques, such as spectrum subtraction and ratioing, are applied to two interrelated spectra measured for a covering nanomaterial and bare substrate to highlight the spectral features related to the nanomaterial. However, using a two-point probe method, the influence from substrate signals can only be weakened rather than completely removed, so the information obtained about a nanomaterial is not quantitative. Indeed, even the influence from substrate signals can be completely removed, electron-based surface analysis techniques face one more problem; that is, strong secondary electron (SE) background at low energies. Regardless of the sample size, the strong intensity of SEs generally observed in spectra measured below 50 eV results in a lack of spectral features because of the SE cascade. Although signals at such low energies should be the best platform to study electron–electron (e–e) interactions in materials and hold great potential to characterize materials[11,12], they are completely buried in the strong SE background. To quantitatively understand e–e interactions and characterize materials, the first step is to extract useful information from the SE signals. However, this step is particularly difficult for a two-point probe method where weak features in core-level signals are identified against background with the naked eye. There is a growing consensus that it is impossible to extract pure nanomaterial information particularly at low energies from just two spectra using simple algebra without any prior knowledge about distinguishing spectral features. Because the two-point probe method cannot provide quantitative information about a nanomaterial, it seems that a method with more probe points, like the four-point probe method, may overcome this limitation. In fact, the feasibility of

this logic has been demonstrated in various fields; for instance, the four-point probe method has been successfully implemented in materials science to precisely determine the electrical resistance of solid-state matter by excluding contributions from parasitic contact resistances[13], and also in radio astronomy as the chop-nod method[14] to detect faint astronomical sources by ground-based telescopes despite the bright, variable sky background. Learning from these successful examples, we realize that the four-point probe method could be a trigger for more efficient use of electron-based surface analysis techniques on nanomaterials.

In this work, we propose the virtual substrate method, which is an extension of the four-point probe method to nanomaterials science, to study substrate-supported nanomaterials without influence from substrate signals even at low energies. Using the virtual substrate method in electron-based surface analysis techniques, the equivalent transmission configuration experiment can be realized from a combination of four interrelated measurements in reflection configuration. Furthermore, uncoupling our reliance on core-level signals, the virtual substrate method enables us to extract information from seemingly featureless spectra. Therefore, the full energy range (white) spectrum, mainly including SEs, can be treated as a useful signal. This perspective is quite different from the established surface analyses that target characteristic peaks in a narrow-range spectrum.

## Results

**The concept of the virtual substrate method.** Although the virtual substrate method is not restricted to surface analysis, the implementation of this principle shown in Fig. 1 is based on surface electron spectroscopy techniques. The raw spectra represent the evolution of a primary electron beam inside a sample driven by the interaction of the sample with moving electrons. From the viewpoint of mathematics, the energy spectrum $J_0(E)$ of a normally incident electron beam can be described by a special vector with one non-zero element representing the incident electron energy. First, a measurement of a bare substrate (substrate A) is considered, where the substrate acts as the scatterer that emits the reflected electrons and SEs. Such a process is essentially a modification of $J_0(E)$, transforming the monochromatic incident electrons into the emitted white electrons. Therefore, the scattering process can be described by the matrix $\mathbf{R}$, and the reflected spectrum from the substrate can be written as $J_{S(A)}(E) = \mathbf{R}J_0(E)$.

Next, we consider the case where a nanomaterial is placed on the top of the substrate, which is also the configuration used for conventional reflection spectroscopy. The electron beam is first incident on the nanomaterial and produces SEs and partially reflected electrons, which can be described by the material-dependent matrix $\mathbf{R^N}$, so the reflection spectrum can be denoted as $\mathbf{R^N}J_0(E)$. In addition to this reflection process, a transmission process also occurs, which is denoted by the material-dependent matrix $\mathbf{T^N}$. These transmitted electrons with spectrum $\mathbf{T^N}J_0(E)$ then interact with the underlying substrate and lead to the reflected spectrum $\mathbf{RT^N}J_0(E)$. These substrate-reflected electrons subsequently pass through the nanomaterial on the top of the substrate, creating the new spectrum $\mathbf{T^NRT^N}J_0(E)$. In this work, we only consider the approximation to the first order; that is, we neglect any further reflection between the nanomaterial and substrate. Furthermore, we can approximate $\mathbf{T^N}$ as unity for high-energy incident electrons (the first $\mathbf{T^N}$ starting from the right in $\mathbf{T^NRT^N}J_0(E)$), which physically corresponds to the complete transmission of high-energy electrons through the ultra-thin nanomaterial. Therefore, the measured spectrum for

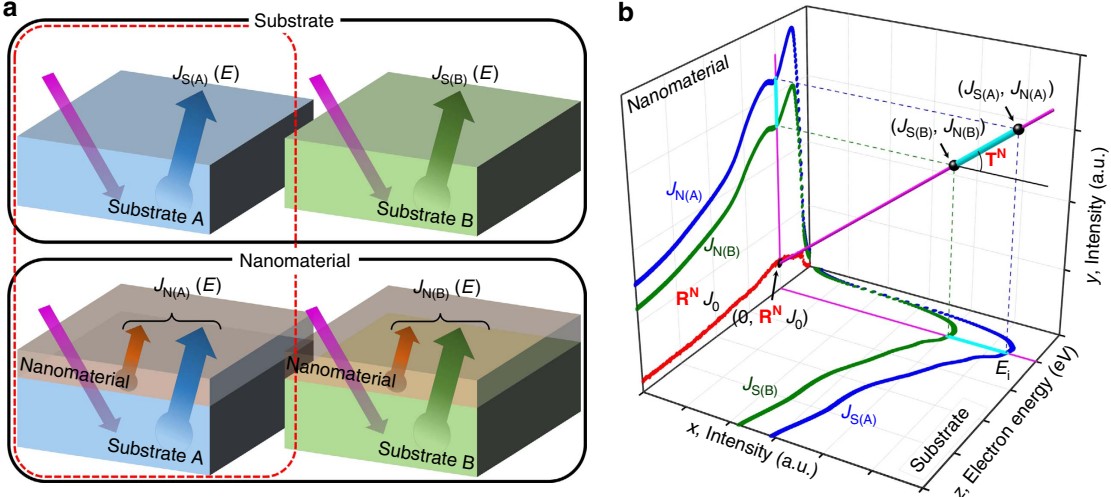

**Figure 1 | Visualization of the virtual substrate method.** (**a**) Schematic diagram of the virtual substrate method implemented in surface analysis. A combination of four interrelated spectra measured for two slightly different bare substrates ($J_{S(A)}$, $J_{S(B)}$) and a target nanomaterial supported on these two substrates ($J_{N(A)}$, $J_{N(B)}$) is used in the virtual substrate method. Different groups defined as 'substrate' and 'nanomaterial' (surrounded by black boxes) are classified from the spectra measured for the bare substrate or nanomaterial. A traditional two-point probe measurement is indicated by a red dashed box. (**b**) Visual representation of the virtual substrate method using a 3D coordinate system, where two spectra obtained using a traditional two-point probe measurement are plotted in pairs orthogonally along the x- and y-axes and share one electron energy axis (z-axis). Two spectra measured for bare substrates (blue and green dots) are plotted in the x–z plane (substrate plane), and the other two spectra measured for the nanomaterial supported on the substrates (blue and green dots) are plotted in the y–z plane (nanomaterial plane). According to the virtual substrate method, the covering nanomaterial information is included in the lines that pass through two points whose x and y coordinates are the intensities of the two spectra in the traditional two-point probe measurements for different substrates at a given energy. The intercept of these lines, $\mathbf{R^N}J_0(E)$, is plotted in the y–z plane (nanomaterial plane) as red dots. One line (purple) at energies $E_i$ is plotted together in the x–z and y–z planes (thin purple lines) along with the deviations in these shallow lines (thick cyan line). At $E_i$, two known points ($J_{S(A)}$, $J_{N(A)}$) and ($J_{S(B)}$, $J_{N(B)}$) obtained by traditional two-point probe measurements with different substrates and the intercept point (0, $\mathbf{R^N}J_0$) are presented as large black dots.

a nanomaterial on a substrate can be written as

$$J_{N(A)}(E) \;=\; \mathbf{R^N}J_0(E) + \mathbf{T^N R}J_0(E). \tag{1}$$

Physically, this means that in a single measurement (that is, conventional reflection measurement), the obtained spectrum for the substrate-supported nanomaterials include contributions from several sources: (i) $\mathbf{R^N}J_0(E)$, the reflection from the nanomaterial and typically SEs originating from the interaction of the high-energy monochromatic incident electrons and the nanomaterial and (ii) $\mathbf{T^N R}J_0(E)$, the transmitted spectrum originating from the substrate-reflected electrons $\mathbf{R}J_0(E)$. This greatly complicates the data processing and prevents extraction of the full information of the target nanomaterials. In a traditional two-point probe measurement (represented by the processes inside the red dashed box in Fig. 1a), measurements are performed on both the substrate and substrate-supported nanomaterial. For the reasons discussed above, the substrate reflection is measured separately and the second term in equation (1) can be written as $\mathbf{T^N}J_{S(A)}(E)$. Thus, we will have

$$J_{N(A)}(E) \;=\; \mathbf{R^N}J_0(E) + \mathbf{T^N}J_{S(A)}(E), \tag{2}$$

where $J_{N(A)}(E)$ and $J_{S(A)}(E)$ are the measured spectra for the substrate-support nanomaterial and bare substrate, respectively. $\mathbf{R^N}$ and $\mathbf{T^N}$ are the reflection and transmission matrices for the nanomaterial, respectively. These matrix elements are quantitatively linked to the e–e interaction. Therefore, by solving $\mathbf{R^N}$ and $\mathbf{T^N}$ using linear equations, we can obtain complete information about the target nanomaterial.

However, even neglecting all off-diagonal elements of $\mathbf{T^N}$ (that is, for an ultrathin nanomaterial such as mono- or bilayer graphene), the number of unknown variables, that is, $r_{i,j}$ ($j = j_0$) for matrix elements of $\mathbf{R^N}$ in a given column referring to primary

incident electron beam energy and $t_{i,j}$ ($i = j$) for matrix elements of $\mathbf{T^N}$ on the principal diagonal, is twice the number of equations in equation (2). That is, there is only one equation with two unknowns at a given energy. Therefore, to obtain a solution, we need an additional set of measurements, which can be obtained by collecting another set of measurements using a different substrate. As shown in Fig. 1a, we then perform the traditional two-point probe measurement with an additional substrate (substrate B) to obtain another system of linear equations, such that

$$J_{N(B)}(E) \;=\; \mathbf{R^N}J_0(E) + \mathbf{T^N}J_{S(B)}(E). \tag{3}$$

With the number of variables now equal to the number of equations (equations (2) and (3)), we can solve the matrices $\mathbf{R^N}$ and $\mathbf{T^N}$ and thereby obtain complete information for a target nanomaterial.

Combining equations (2) and (3) determined according to the four-point probe method, we will have

$$J_{\Delta N}(E) \;=\; \mathbf{T^N}J_{\Delta S}(E), \tag{4}$$

where $J_{\Delta N}(E)$ and $J_{\Delta S}(E)$ are the difference spectra, which can be obtained by subtracting two measured spectra for a substrate-supported nanomaterial ($J_{N(A)}(E)$ and $J_{N(B)}(E)$) and two spectra for the substrates ($J_{S(A)}(E)$ and $J_{S(B)}(E)$). The mathematical expression of $J_{\Delta N}(E)$ is $\mathbf{T^N}(\mathbf{R_A} - \mathbf{R_B})J_0(E)$, while that of $J_{\Delta S}(E)$ is $(\mathbf{R_A} - \mathbf{R_B})J_0(E)$, where $\mathbf{R_A}$ and $\mathbf{R_B}$ are matrix descriptions of the reflection process of substrate A and B, respectively, and can be further simplified as $\mathbf{T^N}\delta\mathbf{R}J_0(E)$ and $\delta\mathbf{R}J_0(E)$, respectively, where $\delta\mathbf{R}$ represents a 'virtual substrate' whose contribution is equivalent to the responses of two substrates to injected electrons ($\delta\mathbf{R} = \mathbf{R_A} - \mathbf{R_B}$). It is obvious that $J_{\Delta S}(E)$ and $J_{\Delta N}(E)$ are the responses of the bare substrate system and

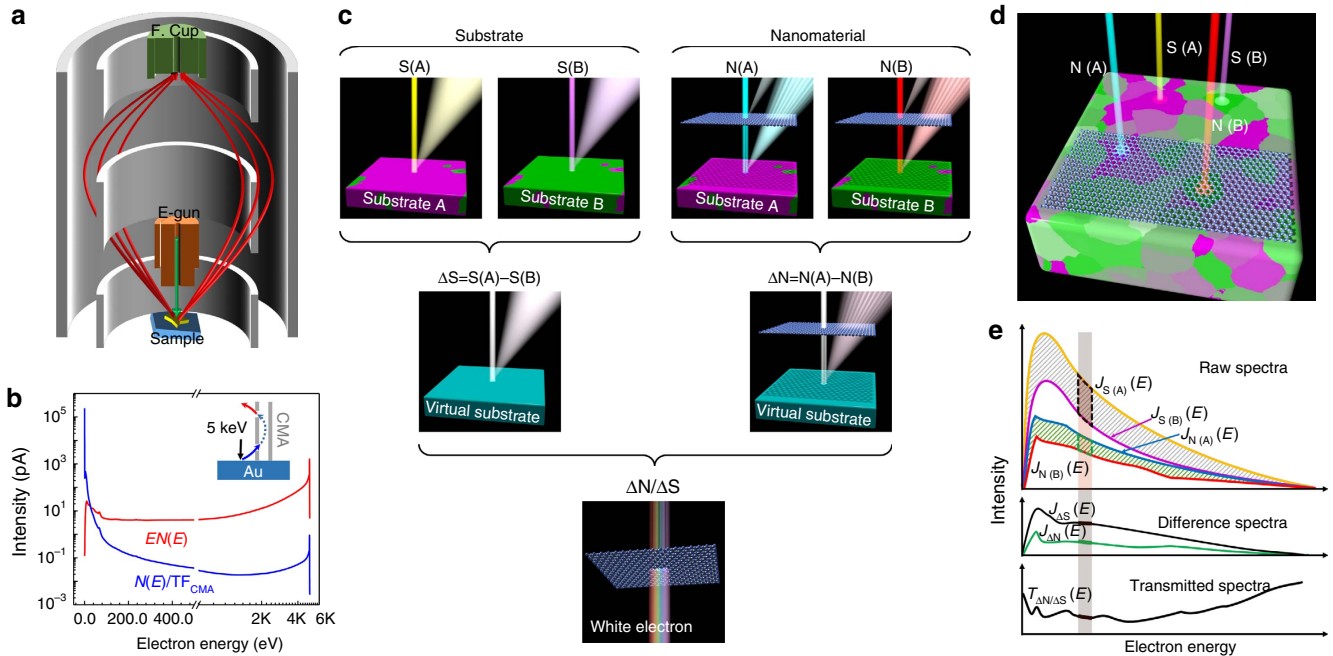

**Figure 2 | Implementation of the virtual substrate method in Auger electron spectroscopy.** (**a**) Experimental setup for AES. (**b**) Raw electron energy spectrum $EN(E)$ (red) and restored spectrum $N(E)/\text{TF}_{\text{CMA}}$ (blue). (**c**) Top: four real experiment configurations, S(A), S(B), N(A) and N(B). Middle: equivalent experiment configurations $\Delta S$ and $\Delta N$ in which virtual substrates (see main text) are used. Bottom: Equivalent experiment configuration $\Delta N/\Delta S$ in which white electrons (see main text) are used as a probe. (**d**) Application of the virtual substrate method to graphene/gold. (**e**) Spectra for the three major steps in a virtual substrate measurement: four raw spectra ($J_{S(A)}$, $J_{S(B)}$, $J_{N(A)}$ and $J_{N(B)}$), two difference spectra ($J_{\Delta S}$ and $J_{\Delta N}$) and one transmitted spectrum ($T_{\Delta N/\Delta S}$).

nanomaterial/substrate system, respectively, to the virtual substrate, which are not related to the concrete substrate used in these systems. The physical meaning of the two difference spectra $J_{\Delta S}(E)$ and $J_{\Delta N}(E)$ then becomes apparent; they are the initial and final states, respectively, for white electrons with the expression $\delta \mathbf{R} J_0(E)$ travelling through a nanomaterial. Therefore, the ratio $J_{\Delta N}(E)/J_{\Delta S}(E)$ (that is, $\mathbf{T}^{\mathbf{N}}\delta\mathbf{R}J_0(E)/\delta\mathbf{R}J_0(E)$) directly reveals the quantitative e–e interaction information for matrix $\mathbf{T}^{\mathbf{N}}$. $\mathbf{T}^{\mathbf{N}}$ is typically a lower triangular matrix that can simply be split into two matrices:

$$
\mathbf{T}^{\mathbf{N}} = \begin{pmatrix} t_{1,1} & & & \\ t_{2,1} & t_{2,2} & & \\ \cdots & \cdots & \ddots & \\ t_{n,1} & t_{n,2} & \cdots & t_{n,n} \end{pmatrix}
$$
$$
= \begin{pmatrix} t_{1,1} & & & \\ & t_{2,2} & & \\ & & \ddots & \\ & & & t_{n,n} \end{pmatrix} + \begin{pmatrix} 0 & & & \\ t_{2,1} & 0 & & \\ \cdots & \cdots & \ddots & \\ t_{n,1} & \cdots & t_{n,n-1} & 0 \end{pmatrix}.
$$
$$(5)$$

The first matrix, which includes only the elements on the principal diagonal, reflects the elastic electron transmission information of the nanomaterial. Elements $t_{i,j}$ $(i=j)$ are the elastic electron transmission of the nanomaterial, which converge to 1 as the index (electron energy) increases. The second matrix includes only those elements $t_{i,j}$ $(i>j)$ below the principal diagonal and is a sparse matrix whose non-zero entries are located in two major regions. One major region is confined to a diagonal band below the main diagonal, providing information about inelastic scattering processes. The lower bandwidth of this region depends on the electron energy and nanomaterial

thickness. The other major region is located far from the main diagonal near the bottom left corner of the matrix and describes the production of SEs in inelastic scattering processes, whose intensities directly reflect the energy loss behaviour of the nanomaterial. For an ultra-thin target nanomaterial, such as mono- or bilayer graphene, the $\mathbf{T}^{\mathbf{N}}$ matrix mainly contains contributions from the first matrix term and can be treated as the elastic electron transmittance of the target nanomaterial. For a thick target nanomaterial, such as few-layer graphene ($n>5$), the second matrix term is dominant, reflecting the accompanying secondary electron emission (SEE) at low energy and providing energy loss information about the target nanomaterial.

In fact, there is another more intuitive way to demonstrate the principle of the virtual substrate method. As shown in Fig. 1b, the relationship between the measured spectra ($J_{S(A)}$, $J_{N(A)}$, $J_{S(B)}$ and $J_{N(B)}$) and the determined elements in the matrices $\mathbf{T}^{\mathbf{N}}$ and $\mathbf{R}^{\mathbf{N}}$ can be visualized as a finite number of lines that pass through the two points ($J_{S(A)}$, $J_{N(A)}$) and ($J_{S(B)}$, $J_{N(B)}$). These intersection points in the lines correspond to every energy in the measured spectra, whose slope and intercept are $\mathbf{T}^{\mathbf{N}}$ and $\mathbf{R}^{\mathbf{N}}J_0(E)$, respectively. According to this relationship, when the intensities of four interrelated raw spectra in the form of the two points ($J_{S(A)}$, $J_{N(A)}$) and ($J_{S(B)}$, $J_{N(B)}$) are considered inputs, then the slope and intercept of the determined lines, $\mathbf{T}^{\mathbf{N}}$ and $\mathbf{R}^{\mathbf{N}}J_0(E)$, respectively, are the outputs, and include only the properties of the nanomaterial. A more intuitive description is that the virtual substrate method converts a line determined from the absolute intensities of four interrelated raw spectra at a given energy in measurable space into a point in parameter space (slope–intercept parameterization of a straight line), where the slope (that is, diagonal elements of $\mathbf{T}^{\mathbf{N}}$) and intercept (that is, $\mathbf{R}^{\mathbf{N}}J_0(E)$) can be considered the equivalent transmitted electron spectrum and equivalent reflected electron spectrum for a free-standing target nanomaterial, respectively. It should be noted that this

'intuitive description' is valid only for thin layers, where it is appropriate to neglect the inelastic contribution.

**White electrons in the virtual substrate method**. Besides removing the influence from substrate signals, implementing the virtual substrate method in electron-based techniques also enables the use of an energy-dispersive full spectrum as a probe to investigate the properties of a target nanomaterial at different energies simultaneously. Figure 2a shows an AES setup with a cylindrical mirror analyser (CMA) including a Faraday cup. Focused electrons are incident on a sample, and emitted electrons are deflected into a CMA, generally with an unclear transmission function ($\mathrm{TF_{CMA}}$), which is accurately measured here[15]. Considering the modification by $\mathrm{TF_{CMA}}$, a measured AES spectrum (gold) can be restored back to the moment just before entering the CMA detector (Fig. 2b). According to this restored spectrum, the number of SEs increases dramatically as the electron energy decreases, which implies that this reflected SE spectrum has the potential to be used as a probe to study substrate-supported nanomaterials. In this method, the nanomaterial is back-illuminated by the reflected SE spectrum as a white electron probe analogous to the widely used white X-rays[16]. But unlike white X-rays, when white electrons are used as a probe, the information carried by the white electrons in the SE energy range is obscured by the undesired electron signals produced by energy loss or the formation of new SEs in inelastic collisions, while white electrons of higher energy travel in a target nanomaterial. Therefore, the energy channels in the measurement are approximately rather than fully separated when the reflected SEs from the underlying substrate are used as a white electron probe. In this case, the underlying substrate acts as a backscatterer of the monochromatic electron beam with fully dispersed energies, forming the white electrons. The transmission of white electrons through the nanomaterial enables the e–e interactions of the nanomaterial supported by the substrate to be studied with ultimate efficiency; however, the initial energy distributions of these white electrons are difficult to measure in reflection configuration because of the influence of the reflection from the nanomaterial. Implementing the virtual substrate method in AES measurements, white electrons originating from substrate-reflected electrons, can be controlled to quantitatively investigate the covering nanomaterial by realizing an equivalent transmission configuration measurement from a combination of four interrelated measurements in reflection configuration, as shown in Fig. 2c. This method uses four interrelated measurements in reflection configuration for the two bare substrate (S(A) and S(B)) and a target nanomaterial supported on two substrates (N(A) and N(B)). Four interrelated spectra, $J_{S(A)}$, $J_{S(B)}$, $J_{N(A)}$ and $J_{N(B)}$, can be obtained from these measurements correspondingly. According to the virtual substrate method, spectrum subtraction is implemented by subtracting two spectra associated with different substrates (substrate A and B) to give a difference spectrum ($J_{\Delta S}(E)$ and $J_{\Delta N}(E)$ for the bare substrate system and nanomaterial/substrate system, respectively), in which an inevitable issue in reflection configuration—SEs excited because of the attenuation of the monochromatic incident electron beam—is completely removed. If there is a virtual substrate whose contribution is equivalent to the deviations of the spectra separately measured on the two different substrates at low energy, $J_{\Delta S}(E)$ and $J_{\Delta N}(E)$ can be viewed as the output from equivalent measurements in reflection mode for the bare virtual substrate ($\Delta S$) and the target nanomaterial supported on the virtual substrate ($\Delta N$), respectively. In this case, $J_{\Delta S}(E)$ containing SEs excited in the virtual substrate and emitted from the surface can be used as a

white electron probe. In contrast, $J_{\Delta N}(E)$ contains an attenuated white electron probe that passes through the nanomaterial together with the accompanying SEs. Therefore, by measuring these two difference spectra, the initial and final states for the white electron travelling through a nanomaterial can be obtained by neglecting the blocking effect of the nanomaterial on the incident electron beam. Furthermore, the transmission information as a function of electron energy is the ratio of the two difference spectra, termed the transmitted spectrum $T_{\Delta N/\Delta S}(E)$, (that is $J_{\Delta N}(E)/J_{\Delta S}(E)$) which can be viewed as the output from an equivalent measurement in transmission configuration ($\Delta N/\Delta S$), in which the white electron is used as a probe to quantitatively investigate nanomaterials based on their subtle changes in the transmission of a nearly transparent nanomaterial. It should be noted that the virtual substrate method is suitable to remove the reflectivity difference between a substrate-supported nanomaterial and bare substrate for most analysis tools in reflection configuration, whereas the white electron probe used to raise efficiency is only applicable to electron-based techniques.

**Practical application of the virtual substrate method**. A practical application of this virtual substrate method using graphene as a representative nanomaterial is shown in Fig. 2d. Here we used a polycrystalline metal substrate composed of alternating micron-sized single-phase grains with different crystallographic orientations instead of different substrates. Generally, the uncertainty range of relative orientations for one type of metal grains should be within 0.5°, while the relative orientation between two types of metal grains should be larger than 4° (Supplementary Fig. 1). A typical virtual substrate method operation involves three major steps (Fig. 2e, Supplementary Figs 2–5). First, four raw spectra are measured by selecting incident positions on different crystallographic orientations in the bare substrate and similar regions covered by graphene sheets. Second, difference–spectra are calculated by subtracting paired spectra with the same experimental configuration. Third, the transmitted spectrum is obtained from the ratio of the two difference spectra. In addition, theoretical approaches are used to remove the disturbance from Auger electrons and accompanying SEs when focusing on transmission information (Supplementary Note 1, Supplementary Figs 6–8). It should be noted that the reliability of these selected incident positions was verified by the consistency of the raw spectra measured at these positions before transferring the target nanomaterial sheets onto half of them, and the relative errors should be within 5%. Atomic force microscopy was used to confirm the absence of wrinkles in the covering nanomaterial layer. The criteria for selecting these measurements points are presented in Supplementary Note 2 and Supplementary Figs 9–13. Generally, short-term repeated measurements for multiple cycles with micrometre distances between different measurement sites were used to minimize the influence of changes in the stability of the instrument over time and sample inhomogeneity.

**Elastic electron transmission**. The virtual substrate method was first investigated using mono- and bilayer graphene. The elastic transmission of mono- and bilayer graphene over the entire energy range is presented in Fig. 3. To confirm the effectiveness of the virtual substrate method, we compared the elastic transmission obtained using different theoretical approaches. The extended Mermin method[17] was used to calculate the electron inelastic mean free path (IMFP) of monolayer graphene from a corresponding energy loss function determined by the WIEN2k package[18]. Using a standard straight-line approximation[19] for the attenuated signal from IMFP only, the elastic transmission $T_n$ of

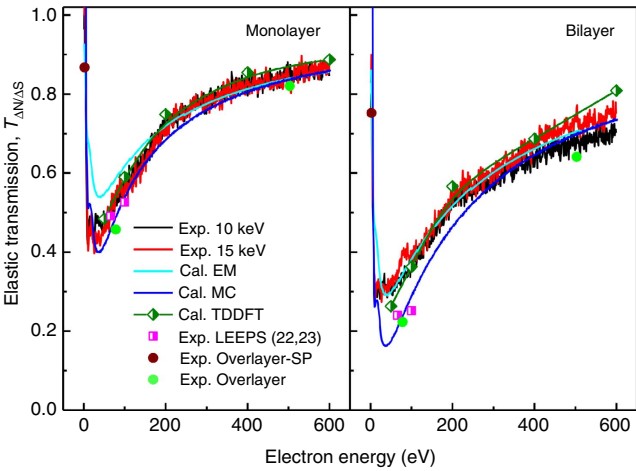

**Figure 3 | Elastic transmission of mono- and bilayer graphene.** Elastic transmission ($T_{\Delta N/\Delta S}(E)$, that is $J_{\Delta N}(E)/J_{\Delta S}(E)$) of monolayer graphene and bilayer graphene measured by the virtual substrate method for primary electron energies of 10 and 15 keV. The elastic transmission was calculated by the extended Mermin (EM), Monte Carlo (MC) and time-dependent density functional theory (TDDFT) methods. The total transmission was measured by low-energy electron point-source (LEEPS) microscopy[22,23] corrected to the elastic transmission by the MC method. The elastic transmission was measured by the overlayer method for graphene/gold at 2.3 eV, associated with gold surface plasmons (overlayer-SP), and for graphene/$SiO_2$ samples at 78 and 503 eV (overlayer).

$n$-layer graphene can be estimated by

$$T_n = \exp(-nd_0/\lambda_{IMFP} \times \cos\theta), \qquad (6)$$

where $d_0$ is the thickness of graphene (0.335 nm), $\lambda_{IMFP}$ is the IMFP of monolayer graphene and $\theta$ is the emission angle. The results obtained by this method agree well with those obtained from the virtual substrate measurements, except for the energy range of 10–200 eV because of the lack of a multiple scattering effect. For bilayer graphene, excellent agreement is achieved over the entire energy range because of the compensation of the diffraction effect, mainly provided by the errors introduced by using the dielectric function of the well-known jellium model to describe the electrical properties of single-crystal graphene. Using the Monte Carlo (MC) method[20], the elastic interactions of electrons with carbon atoms are predicted by considering the zigzag trajectory of electrons inside graphene. The MC method shows excellent agreement with the virtual substrate measurements over the entire energy range for monolayer graphene, with some deviation from 10 to 300 eV for bilayer graphene. Time-dependent density functional theory (TDDFT)[21] calculations were performed to include the influence of the two-dimensional crystal. The elastic transmission predicted by TDDFT corresponded well with the virtual substrate measurements and theoretical predictions of the MC method for monolayer graphene and exhibited a deviation of ~15% at high energies (>300 eV) for bilayer graphene. Other experimental techniques were also used to measure elastic transmission. The virtual substrate measurements show excellent agreement with existing electron point source microscopy at 66 eV[22] and 100 eV[23] and with elastic transmission measurements performed in this work using the AES overlayer method[24] at 2.3 eV (gold surface plasmon), 78 eV (Si LVV Auger transition) and 503 eV (O KLL Auger transition). It should be noted that the data point at 2.3 eV is the attenuation of surface plasmons of gold (approximately equalling the attenuation of electrons) by graphene sheets, which was estimated from the

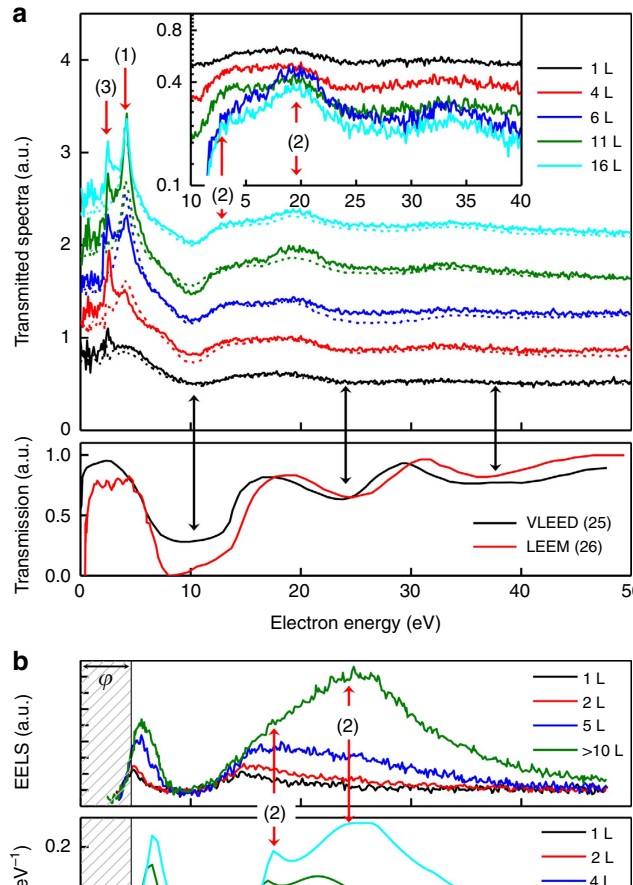

**Figure 4 | Material characterization using the virtual substrate method.** (**a**) Top: transmitted spectra for one-, four-, six-, 11- and 14-layer graphene for primary electron energies of 10 keV (dashed lines) and 15 keV (solid lines). Insets show the transmitted spectra at 15 keV without an offset. Bottom: electron transmission spectrum ($T_{VLEED}$)[25] measured by very-low-energy electron diffraction (VLEED) and transmission data ($T_{LEEM}$)[26] estimated from the reflectivity spectra ($T_{LEEM} = 1 - R_{LEEM}$) obtained from low-energy electron microscopy (LEEM). (**b**) Top: electron energy-loss spectra (EELS) for one-, two-, five- and several-layer graphene showing $\pi$ and $\pi + \sigma$ plasmons[27]. Bottom: differential surface excitation parameter (DSEP) spectra for the few-layer graphene/gold system. The shaded region is the energy loss below the work function of graphite ($\varphi = 4.6$ eV). For straight comparison, the work function of graphite relative to the cylindrical mirror analyser was determined a priori.

intensities of surface plasmon gain peaks observed at a surface plasmon energy above the vacuum level in difference spectra for $n$-layer graphene/gold ($n = 4, 6, 11$ and 14) with an incident electron energy of 10 keV (Supplementary Fig. 5d).

**Characterization of nanomaterials.** For few-layer graphene sheets, accompanying SE features appear in the transmitted spectra (Fig. 4a) together with the transmitted spectra roughly estimated by very-low-energy electron diffraction (VLEED) for single-crystal graphite[25] and low-energy electron microscopy (LEEM) for eight-layer graphene on SiC (ref. 26). Although transmission data from VLEED and LEEM are related to the

surface properties of graphite and graphene rather than the overall properties of graphene like those obtained by the virtual substrate method, strong, consistent fluctuations of electron energy caused by the diffraction of the crystal potential at certain energies are observed in all the transmitted spectra. Peaks at 2.3 and 4 eV and the plateau at 12–20 eV (highlighted by red arrows (3), (1) and (2), respectively) appear only in the virtual substrate measurements. All of these features become more pronounced as the number of sheets increases because of the specific accompanying SE contribution; however, they originate from different mechanisms. For instance, peak (1) relates to the $\sigma$–$\sigma^*$ transition in graphene. The increase in peak height and sharpness with sheet number reflects the competition of cascade SE peaks between graphene and the underlying gold substrate (a broad plateau at approximately 8 eV). The plateau structure (2) is associated with $\pi + \sigma$ plasmon excitation in graphene layers. To prove this association, the plasmon spectrum of a free-standing graphene film measured by electron energy-loss spectroscopy (EELS)[27] and the theoretical prediction of the differential surface excitation parameter (DSEP), including coupling excitation with the underlying substrate, are presented in Fig. 4b. The plateau structure in the virtual substrate measurements occurs at the exact energies corresponding to the $\pi + \sigma$ plasmon energies observed in the EELS and DSEP spectra minus the work function $\varphi$ of graphene sheets (4.6 eV[28]); $\varphi$ of the sample with respect to the CMA was already considered to determine the onset of the spectra. This result demonstrates that plasmons excited by electron energy-loss decay via the generation of single electron–hole pairs act as a source of SEs. No features related to $\pi$ plasmon excitation are found, indicating that $\pi$ plasmon decay does not contribute to SEE. Peak (3), assigned to the gain of a surface plasmon quantum of gold, is caused by emitted SEs that gained a surface plasmon quantum in the effective surface plasmon area after overcoming $\varphi$ as the reverse reaction of supersurface electron scattering[29].

## Discussion

Although multi-point probe measurements have been used to obtain information from measured spectra for many years already, like multi-spectral approaches using Auger electron microscopy[30], the concept behind the virtual substrate measurement presented in this work is a new development. In fact, besides the accompanying SE features in transmitted spectra, the electronic properties of a target nanomaterial determined by a virtual substrate measurement can also be used as a descriptor for nanomaterial characterization. To demonstrate this, virtual substrate measurements were performed for investigating mono- and bilayer molybdenum disulfide ($MoS_2$) samples, which have different electronic properties in the low energy range as identified by optical spectroscopy[31]. The determined transmitted spectra for mono- and bilayer $MoS_2$ are presented in Fig. 5. Consistent fluctuation of electron energy caused by a combination of inelastic scattering, accompanying SE and the diffraction effect is observed in the transmitted spectra for mono- and bilayer $MoS_2$ over the whole energy range, except for between 2 eV and 5 eV, as highlighted by blue arrows. When the electron energy exceeds 2 eV, the intensity of transmitted spectra for monolayer $MoS_2$ decreases gradually with increasing electron energy until the energy reaches about 5 eV. For bilayer $MoS_2$, the intensity of transmitted spectra decline sharply until this behaviour suddenly stops at about 2.5 eV. This variation between mono- and bilayer $MoS_2$ may be caused by different e–e interactions near the band gap energy, different out-of-plane properties, or different interfacial properties when mono- and bilayer $MoS_2$ contact with the gold substrate (these mechanisms

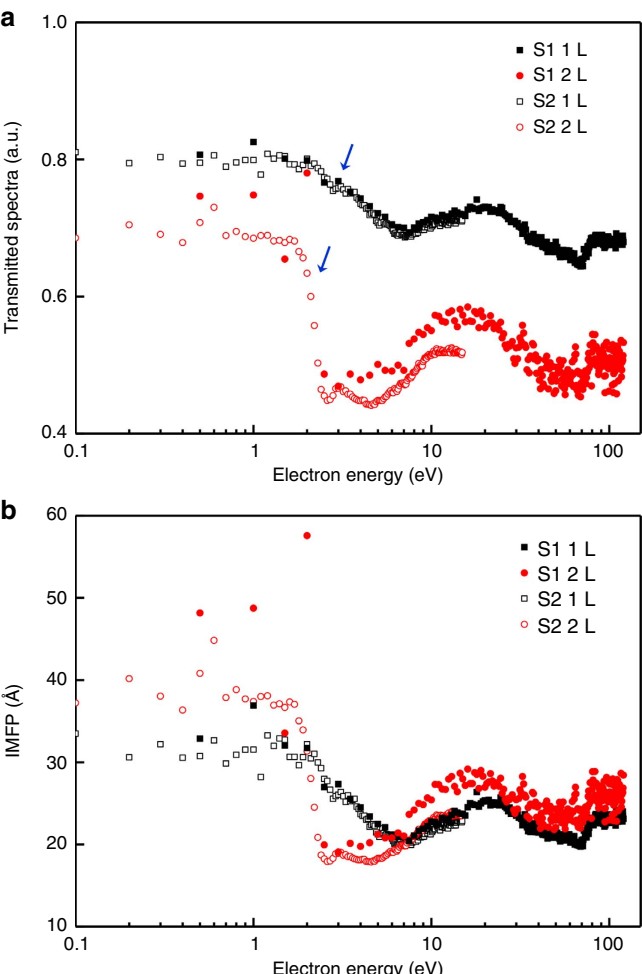

**Figure 5 | Electronic properties of monolayer and bilayer MoS₂.**
(**a**) Transmitted spectra for mono- and bilayer $MoS_2$ obtained at a primary electron energy of 20 keV. Two different $MoS_2$ sheets supported on the same batch of substrates, denoted as S1 and S2, were used in virtual substrate measurements with energy ranges of up to 12 eV in 0.1 eV increments and 120 eV in 0.5 eV increments, respectively. (**b**) Corresponding inelastic mean free path (IMFP), $\lambda_{IMFP}$, of mono- and bilayer $MoS_2$ for a primary electron energy of 20 keV.

will be discussed elsewhere). Such variation in measured transmitted spectra can be used as an indicator in electron-beam techniques to distinguish substrate-supported mono- and bilayer $MoS_2$. Furthermore, the differences between mono- and bilayer $MoS_2$ are more obvious when organizing these transmitted spectra in the form of IMFP by reversing equation (1); that is, $\lambda_{IMFP} = -(nd_0)/(\ln T_n \times \cos \theta)$. The determined IMFPs for mono- and bilayer $MoS_2$ agreed well over the whole energy range, except for between 2 eV and 5 eV, which is consistent with the observed transmitted spectra.

The experimental conditions required for a virtual substrate measurement are similar to those of traditional measurements except for the substrate on which the target nanomaterial is supported. A tailor-made substrate is important in the virtual substrate method; in this work, the substrate is polycrystalline gold. In fact, strict requirements for the substrate are only necessary for highly precise, high-speed quantitative studies of target nanomaterials. To obtain low-precision measurements, virtual substrate measurements can be performed using almost any two substrates with different element composition, surface morphology, and crystal quality. However, polycrystalline metal

substrates are the best choice for highly precise, high-speed quantitative studies of target nanomaterials because of their similar surface barrier and degree of interaction with the covering nanomaterial. For instance, a hole-patterned SiO$_2$ substrate has also been used to investigate multilayer graphene by virtual substrate measurements. Four spectra measured for flat and hole regions of the SiO$_2$ substrate without and with a covering graphene layer are used in this measurement instead of spectra measured for two different types of gold grains. The results of this measurement were broadly in line with those obtained for a graphene/polycrystalline gold system, but had very low precision. This demonstrates that we are able to design other tailor-made substrates in accordance with specific conditions; however, tailor-made substrates that can produce nearly identical reflected SE spectra at two determined measurement points are necessary to perform virtual substrate measurement.

For research purposes, the target nanomaterial can be transferred onto the tailor-made substrate; however, for industrial purposes, it is usual to study a target nanomaterial on a given substrate (any arbitrary substrate). Stringent demands for the substrate are essential to extract pure information of target nanomaterials with high precision through virtual substrate measurements. These stringent demands for the substrate are bound to restrict the scope of this technique in wider applications, especially industrial ones. However, there are exceptions even in industry; for instance, the virtual substrate method will be a very efficient tool to quantitatively investigate passive films on stainless steel, which is a typical nanomaterial/polycrystalline substrate system. In fact, there is another way to implement this virtual substrate method by which almost any given nanomaterial/ substrate combination with a substrate that is not completely uniform, like a single crystal, can be investigated just from a 'monochromatic' SEM image without any designed substrate, and even without selecting measurement points. In this spectral imaging approach, the pixels of a SEM image formed by detected electrons at a given energy are considered as pixel sized 'fictitious grains' and used to perform the virtual substrate measurement as described in Supplementary Note 3 and Supplementary Fig. 14. This new approach has allowed us to study a target nanomaterial on a given substrate as long as it displays fairly stable intensity distributions in a SEM image. A limitation of this approach is the huge amount of data generated during measurement; however, it does represent a possible future direction for the virtual substrate method.

In summary, the virtual substrate method represents a benchmark to provide 'free-standing' nanomaterial information from measurements of substrate-supported samples, which, in principle, can be easily implemented in many more reflection-configuration techniques than surface analysis techniques and does not demand extra investment in equipment. Implemented in electron-based surface analysis techniques, this method expands the energy scale of analysis down to several electron volts and thus allows one to quantitatively probe the e–e interactions of a nanomaterial and observe 'hidden' electronic energy transfer to and from a nanomaterial on a substrate, which is visualized as emitted SE features in equivalent 'transmitted' spectra. Furthermore, using ordinary SE signals, the virtual substrate method outrivals conventional methods based on core-level signals in signal-to-background ratio by orders of magnitude. Thus, the virtual substrate method holds great potential for manufacture monitoring and quality control.

## Methods

**Substrate preparation.** Gold layers were evaporated on Si (100) substrates with thin titanium buffer layers using electron-beam evaporation (RDEB-1206K, R-DEC Co. Ltd., Ibaraki, Japan), as shown in Supplementary Fig. 1. The thicknesses of the

titanium and gold metal layers were 5.0 and 200.0 nm, respectively, and they were deposited at rates of 0.05 and 0.2 nm s$^{-1}$, respectively. The chamber pressure was $\sim 1.0 \times 10^{-5}$ Pa. After evaporation, the samples were annealed by rapid thermal annealing (QHC-P410, ULVAC-RIKO Inc., Kanagawa, Japan) under a N$_2$ atmosphere at 300 °C for 30 s.

**Graphene fabrication.** Graphene flakes were produced on the gold substrates by mechanical exfoliation[32] as shown in Supplementary Figs 2 and 4. The number of graphene layers was estimated by atomic force microscopy and further confirmed by Raman spectroscopy, particularly for mono- and bilayer graphene[33].

**Virtual substrate measurement.** The raw spectra in the virtual substrate measurements were measured at room temperature with a scanning Auger electron spectroscope (SAM650, ULVAC-PHI, Kanagawa, Japan) with a CMA (Supplementary Figs 3 and 5). The take-off angle of the instrument was 42.3 ± 6°. The incident electron beam current for these raw spectra was $\sim 0.87$ nA, as calibrated with a Faraday cup before the measurements. The raw spectra were averaged from eight different sample regions ($\sim 490$ nm$^2$) on the bare substrate as well as on graphene samples with different numbers of layers.

**Condition number of the measurement.** From the error propagation analysis of the expression of the transmitted spectrum $T_{\Delta N/\Delta S} = J_{\Delta N}/J_{\Delta S}$ in a virtual substrate measurement, the relationship between the relative errors in the raw spectra as the input and the transmitted spectrum as the output can be simply obtained as

$$\frac{\Delta T_{\Delta N/\Delta S}}{T_{\Delta N/\Delta S}} = \frac{J_{S(A)}}{J_{\Delta S}}\frac{\Delta J_{S(A)}}{J_{S(A)}} + \frac{J_{S(B)}}{J_{\Delta S}}\frac{\Delta J_{S(B)}}{J_{S(B)}} + \frac{J_{N(A)}}{J_{\Delta N}}\frac{\Delta J_{N(A)}}{J_{N(A)}} + \frac{J_{N(B)}}{J_{\Delta N}}\frac{\Delta J_{N(B)}}{J_{N(B)}}, \quad (7)$$

where $\Delta J_{S(A)}$, $\Delta J_{S(B)}$, $\Delta J_{N(A)}$ and $\Delta J_{N(B)}$ are small given changes in the raw spectra and $\Delta T_{\Delta N/\Delta S}$ is the resulting change in the transmitted spectrum. The relative errors in the raw spectra, such as $\Delta J_{S(A)}/J_{S(A)}$, $\Delta J_{S(B)}/J_{S(B)}$, $\Delta J_{N(A)}/J_{N(A)}$ and $\Delta J_{N(B)}/J_{N(B)}$, are enhanced $J_{S(A)}/J_{\Delta S}$, $J_{S(B)}/J_{\Delta S}$, $J_{N(A)}/J_{\Delta N}$ and $J_{N(B)}/J_{\Delta N}$ times in the transmitted spectrum $T_{\Delta N/\Delta S}$, respectively. The condition number of the virtual substrate measurement for the initial error in a specific raw spectrum equals the ratio of this raw spectrum to the corresponding difference spectrum; for instance, $J_{S(A)}/J_{\Delta S}$, $J_{S(B)}/J_{\Delta S}$, $J_{N(A)}/J_{\Delta N}$ and $J_{N(B)}/J_{\Delta N}$ are the condition numbers of this technique for the initial errors in $J_{S(A)}$, $J_{S(B)}$, $J_{N(A)}$ and $J_{N(B)}$, respectively. To study mono- and bilayer graphene, the condition numbers of the virtual substrate measurement using the presented tailor-made polycrystalline gold substrate (with a relative orientation of $\sim 4°$ between the two types of gold grains, as shown in Supplementary Fig. 1) have similar values regardless of the initial errors in the raw spectra measured on the bare substrate or covering graphene. The condition number is $\sim 15$ for an energy level $<100$ eV and $\sim 10$ in the energy range of 100–600 eV. Although the virtual substrate measurement is an ill-conditioned system with a high condition number, the condition number can be artificially reduced to 4–6 by using two gold grains with larger relative orientations of 10–12°.

**Transmission calculation.** The MC and TDDFT methods were used to calculate elastic electron transmission without any adjusted parameters. In the MC calculation[34], the elastic scattering was determined by the Mott cross section based on the muffin-tin model potential, and the inelastic scattering was determined by the extended Mermin method, whose only input, the energy-loss function, was provided by the WIEN2k package. Furthermore, a quantum dynamic TDDFT calculation[35] that fully accounted for the carbon atoms of the target graphene and elastic/inelastic electron scattering was carried out for the same purpose. The electron transmission coefficient was calculated from the ratio of the time-averaged transmitted current to incident current, and the elastic component was derived using the MC method, in which the proportion of elastic electrons was provided.

**Theoretical modification.** Theoretical approaches were used to purify the elastic transmission information for mono- and bilayer graphene in the virtual substrate measurements by considering the contributions from Auger electron emission, inelastic scattering and accompanying SEE (Supplementary Fig. 8). The Auger electron contribution was removed from the transmitted spectra by subtracting scaled-down Auger peaks detected in the raw spectra. The inelastic scattering process and accompanying SEE contributions were removed using a self-adaptive iterative MC simulation programme that only exists when white electrons are used as a probe in electron-based techniques, as discussed in Supplementary Note 1. The contribution of the surface potential barrier gap between the nanomaterial and substrate was removed using a square barrier model, where the constant electronic potential in the interior of the sample was defined as the sum of the kinetic energy at the Fermi level (9.0 eV for gold and 20.2 eV for graphene) and $\varphi$ of the material (5.1 eV for gold and 4.2 for graphene).

**Data availability.** The data sets generated during and/or analysed during the current study are available from the corresponding author on reasonable request.

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

## Acknowledgements

We thank Professor K. Goto, Professor W.S.M. Werner, Professor M.S. Xu, Professor M.M. Ma, Dr S.F. Mao, Dr Y.G. Li, Dr R.G. Zeng, Dr J. Hu, Dr C. Chen and Dr L. Sun for helpful comments and discussions. This research was partially supported by a Grant-in-Aid (JSPS KAKENHI Grant Number JP25107004) from the Japan Society for the Promotion of Science (JSPS), a Grant-in-Aid (JSPS KAKENHI Grant Number JP25400409) from MEXT and the National Natural Science Foundation of China (No. 11574289). The numerical TDDFT calculations were performed on supercomputers at the Institute for Solid State Physics, University of Tokyo and the Research Center for Computational Science, National Institutes of Natural Sciences Okazaki Research Facilities, Japan.

## Author contributions

B.D., H.Y. and S.T. supervised the project. B.D. designed the research and wrote the manuscript with important input from all authors. B.D., J.L., M.Y. and K.T. performed the experiments. B.D., Y.U., K.W. and N.T.C. performed the calculations. All authors discussed the results and commented on the manuscript.

## Additional information

**Competing interests:** The authors declare no competing financial interests.

