## [Peer Review File · Nature Communications]

Reviewers' comments:

Reviewer #1 (Remarks to the Author):

This paper covers an interesting area with thorough work to measure certain properties of multilayer graphene on a gold substrate, in vacuo, by developing an electron based method in which the interfering effect of the substrate is effectively removed by a structured set of spectral differences. Electron beam methods have excellent spatial resolution. There are some major and minor points that require addressing:

Major points

1) The title to the work is unhelpful and misleading. It would only be "universal" if one could apply it universally to all materials or to one material to get universal information in some way. Can we apply it to insulators? Can we apply it to organic materials – even conducting organic materials which the beam would fry? None of these is the case for the reported experiments. Nor does it "quantify nanomaterials". It would if we could distinguish 13-layer graphene from 14-layer graphene but that is not the case. Raman does allow such analysis (as described in this work). This paper is, as presented, a "vacuum-based electron beam method that provides characterisation of nanomaterials by removing underlying substrate contributions". The title needs to be changed to fit the content of the paper.

2) The use of the phrase "universal method, named the nano-chop-nod method", in the abstract, is unhelpful. This is basically spectrum subtraction. Yes spectrum subtraction may be universally used where the resulting data concerns linear summations but this is not proved for the first time here, it has been used for decades! The reported study may be based on the chop-nod method of infra-red astronomy but the translation of the method from one field to another is not an exact parallel nor is the phraseology very helpful here. In astronomy, the chop part is no different from the AC modulation methods used in the 1970s to "see" AES signals – it improves the visibility by improving the signal-to-background ratio, rarely the signal-to-random (i.e shot) noise. The nod part in infra-red astronomy allows a better fit to the background at the signal. The above phrase should be replaced by "electron scattering method, based on the spectral difference concept of the chop-nod method used in infra-red astronomy," so that the sentence reads "Here, we propose an electron scattering method, based on the spectral difference concept of the chop-nod method used in infra-red astronomy, to characterize nanomaterials without the influence of underlying substrate signals"

3) The sentence that follows it, "This approach was inspired by the chop-nod method in radio astronomy, which is used to measure the universe", is pretentious and should be deleted. The authors should be aware that this is basically a sum and difference approach assuming that linear additions and subtractions generate the required result. Such sum and difference approaches are not at all new in multi-spectral approaches. See, for example Barkshire et al 25 years ago in Surface and Interface Analysis 17 (1991) 203-208. They again mix 4 spectra but now recorded simultaneously and to remove topography.

4) There is an unfortunate usage of English where grandiose words such as "universal", "universe" and numbers "20 orders of magnitude" are spread like confetti without justification. Such words should be deleted since they are misleadingly used.

Minor points

5) There is general confusion in the text whether the phrase "nano-chop-nod method" refers to the electron based method, as described in the paper, or a more general concept of the basic acquisition of spectra with and without X that are assumed to be linear additions of effects that may then be removed by the use of simple algebra. That concept is demonstrated in a particular implementation here.

6) The explanation of the chop-nod method on page 3 needs significant improvement.

7) Page 3, line 50, the text erroneously states that the signal-to-noise is improved. The random noise cannot be reduced by the procedure, indeed, by the subtraction it is increased here by approximately by 1.4. The background may be removed increasing the signal-to-background ratio. Maybe this is

what the authors were thinking. The text requires correction.

8) Page 3 line 53 the word "which" refers back to the word "signals". This is clearly nonsense. What the authors may refer to is the dimensional size being 20 orders of magnitude smaller. This clause should be deleted as irrelevant.

9) The ordinate labelled "elastic transmission" in Figure 2 is unclear. Either here or in the caption, the algebraic symbol defined in the text should be given – i.e. how this relates to $J\Delta A$, etc.

10) In the method on page 18, it is noted that similar regions are used on both the bare and graphene-covered substrates. How does the user know when the chosen regions are suitably similar. What defines this, both the experimental choice of the grain in the instrument when conducting the measurements and in terms of the number of degrees of crystal mis-orientation involved. This work requires major revision.

Reviewer #2 (Remarks to the Author):

The authors have recognized that a technique from astronomy can be adapted to separate the signal from a nanomaterial and its underlying substrate. The authors show that the adaptation to electron spectroscopies is not trivial, but can be very useful as a general method for studying supported nanomaterials.

I really like the concepts behind this work, but found a number of shortcomings. The first is simply that, in its present form, the work doesn't excite one as much as it could. A second issue (perhaps the cause of the first) is that some of the concepts are not clear in the main text. It is only in the Methods section that it is shown why one samples the substrate and substrate+nanomaterial for at least two different substrate types (4 spectra), i.e., this removes the reflectivity difference between substrate and nanomaterial. Figure 1 is quite complex, but never really makes this point.

I don't think that this is a Nature Communications in its present form. However, I like the idea and it would be valuable to have it disseminated widely. Perhaps this can be reconsidered after extensive edits to clarify the concepts. Some of this might involve moving material from Methods to the main text (and a large portion of Methods probably should move to Supplemental).

Here are some specific questions and comments, keyed to the manuscript line numbers and figures:

L41: There is always a surface contribution because surface/volume ratio is large in nanomaterials. It only becomes apparent much later (in Methods), that it's the surface reflection that is the issue.

Fig. 2: Are the LEEPS results obtained by the authors or from the literature? The way the caption is written and referenced makes it sound as though the authors did the measurements; however, this is a fairly specialized measurement, so it seems likely that the result is from the literature. If these are literature results, the figure label should clearly state that the results are from Refs. 23,24.

Fig. 3: Same issue as Fig. 2, with regard to VLEED and LEEM results. It is essential to make clear whether you performed the measurement based on the techniques described in the reference, or took the actual results from the reference.

L131: I couldn't figure out the phrase "biased estimated thickness"; what biases the estimate?

Reviewer #3 (Remarks to the Author):

This manuscript describes an interesting method to extract fine information from secondary electron emission through a nanomaterial film on substrate. The so-called chop-nod method uses the low energy secondary electron as "white electron probe". As an example the authors show the chop-nod spectrum from graphene on Au substrate, which is far more informative than normal AES. The idea of this chop-nod method is very new and interesting, and useful for study of nanomaterials. I therefore supports the publication of this manuscript in Nature Communication.

The method is however also quite restricted to specific conditions. For example one needs two different substrates in order to obtain a difference spectrum. However growing the same nanomaterial on different substrates would in most case result in films of different structures as well as different thickness. Identical films on different substrate like graphene on Au is not generally available, which limits the applicability of this method.

Responses to the first referee's comments

Comment 1 (Major point 1): *The title to the work is unhelpful and misleading. It would only be “universal” if one could apply it universally to all materials or to one material to get universal information in some way. Can we apply it to insulators? Can we apply it to organic materials – even conducting organic materials which the beam would fry? None of these is the case for the reported experiments. Nor does it “quantify nanomaterials”. It would if we could distinguish 13-layer graphene from 14-layer graphene but that is not the case. Raman does allow such analysis (as described in this work). This paper is, as presented, a “vacuum-based electron beam method that provides characterisation of nanomaterials by removing underlying substrate contributions”. The title needs to be changed to fit the content of the paper.*

Response to Comment 1: We agree that the previous title did not accurately describe our method. We have revised the title to “Virtual substrate method for nanomaterials characterization”.

Comment 2: (Major point 2) *The use of the phrase “universal method, named the nano-chop-nod method”, in the abstract, is unhelpful. This is basically spectrum subtraction. Yes spectrum subtraction may be universally used where the resulting data concerns linear summations but this is not proved for the first time here, it has been used for decades! The reported study may be based on the chop-nod method of infra-red astronomy but the translation of the method from one field to another is not an exact parallel nor is the phraseology very helpful here. In astronomy, the chop part is no different from the AC modulation methods used in the 1970s to “see” AES signals – it improves the visibility by improving the signal-to-background ratio, rarely the signal-to-random (i.e shot) noise. The nod part in infra-red astronomy allows a better fit to the background at the signal. The above phrase should be replaced by “electron scattering method, based on the spectral difference concept of the chop-nod method used in infra-red astronomy,” so that the sentence reads “Here, we propose an electron scattering method, based on the spectral difference concept of the chop-nod method used in infra-red astronomy, to characterize nanomaterials without the influence of underlying substrate signals”*

Response to Comment 2: We agree that “universal” is an unsuitable term to describe the present method. We have deleted the sentence containing the term “universal”. According to the referee’s suggestion, the following new sentence was included on line 21 “**Here, we propose a virtual substrate method, inspired by the four-point probe technique for resistance measurement as well as the chop-nod method in infrared astronomy, to characterize nanomaterials without the influence of underlying substrate signals from four interrelated measurements.**” We use the term “virtual substrate method” in the revised manuscript instead of “nano-chop-nod method”.

To demonstrate why we chose virtual substrate method as the new name for the presented technology, we have added the following text on page 7 and 8 in the results section.

- “The mathematical expression of $J_{\Delta N}(E)$ is $\mathbf{T}^N(\mathbf{R}_A - \mathbf{R}_B)J_0(E)$, while that of $J_{\Delta S}(E)$ is $(\mathbf{R}_A - \mathbf{R}_B)J_0(E)$, where \mathbf{R}_A and \mathbf{R}_B are matrix descriptions of the reflection process of substrate A and B, respectively, and can be further simplified as $\delta\mathbf{R}J_0(E)$ and $\mathbf{T}^N\delta\mathbf{R}J_0(E)$, respectively, where $\delta\mathbf{R}$ represents a “virtual substrate” whose contribution is equivalent to the responses of two substrates to injected electrons ($\delta\mathbf{R} = \mathbf{R}_A - \mathbf{R}_B$). It is obvious that $J_{\Delta S}(E)$ and $J_{\Delta N}(E)$ are the responses of the bare substrate system and nanomaterial/substrate system, respectively, to the virtual substrate, which are not related to the concrete substrate used in these systems.”

Comment 3: (Major points 3) *The sentence that follows it, “This approach was inspired by the chop-nod method in radio astronomy, which is used to measure the universe”, is pretentious and should be deleted. The authors should be aware that this is basically a sum and difference approach assuming that linear additions and subtractions generate the required result. Such sum and difference approaches are not at all new in multi-spectral approaches. See, for example Barkshire et al 25 years ago in Surface and Interface Analysis 17 (1991) 203-208. They again mix 4 spectra but now recorded simultaneously and to remove topography.*

Response to Comment 3: We agree with you that the sentence “This approach was inspired by the chop-nod method in radio astronomy, which is used to measure the universe” is inappropriate. We have removed this sentence and replaced it with a new one to highlight our new concept of white electrons. The following sentence was added on line 24, “Implementing this method in secondary electron (SE) microscopy, a trackable SE spectrum associated with the reflectivity difference between two different substrates can be controlled. The trackable SE spectrum is used to quantitatively investigate the covering nanomaterial based on subtle changes in the transmission of the nanomaterial with extremely high efficiency rivaling that of conventional core-level electrons.”

To demonstrate the essential difference between the present virtual substrate method and standard data treatment, we have added the following paragraph to the discussion section on page 15 and 16.

- “Although multi-point probe measurements have been used to obtain information from measured spectra for many years already, like multi-spectral approaches using Auger electron microscopy [30], regardless of how many interrelated spectra are involved, the concept behind the virtual substrate measurement presented in this work is quite different from the standard data treatment methods represented by subtraction and ratioing techniques. Firstly, the essence of standard data treatment is to screen out useful data points from measured spectra, and draw a conclusion according to the information obtained from these selected data points. In contrast, the essence of the virtual substrate method is to make “useless” data points become useful. That is, with the help of the virtual substrate method, data points that are meaningless alone in individual measurements can be converted into useful data points with direct physical meaning. This

breaks out from the old pattern of thinking that useless and useful data points are completely isolated from each other. Secondly, creating trackable white electrons from the interrelation between absolute intensities of interrelated measured spectra instead of expecting a realistic signal in a measured spectrum (relative intensities in one spectrum) is a completely new approach in surface analysis that helps data treatment to move away from depending on identifying signal features with the naked eye.”

Accordingly, we have added a new Ref. 30 to the manuscript:

[30] Barkshire, I. R. *et al.* Topographical contrast using scatter diagrams and correlated images from four backscattered electron detectors. *Surface Interface Analysis* **17**, 203 (1991).

Comment 4: (Major points 4) *There is an unfortunate usage of English where grandiose words such as “universal”, “universe” and numbers “20 orders of magnitude” are spread like confetti without justification. Such words should be deleted since they are misleadingly used.*

Response to Comment 4: We apologize for the inappropriate use of the phrases “universal”, “universe” and “20 orders of magnitude”. We have deleted these in the revised manuscript.

Minor points

Comment 5: *There is general confusion in the text whether the phrase “nano-chop-nod method” refers to the electron based method, as described in the paper, or a more general concept of the basic acquisition of spectra with and without X that are assumed to be linear additions of effects that may then be removed by the use of simple algebra. That concept is demonstrated in a particular implementation here.*

Response to Comment 5: We apologize for our misguided description about the “nano-chop-nod” method. In the revised manuscript, we use the term “four-point probe method” to refer to the more general concept of the basic acquisition of spectra with and without X that are assumed to be linear additions of effects that may then be removed by the use of simple algebra. In this case, the well-known four-point probe resistance measurement technique in materials science as well as the chop-nod method for detecting faint astronomical sources in radio astronomy can be considered as successful applications of the four-point probe method in different fields. Therefore, the present virtual substrate method (called the nano-chop-nod method in the original manuscript) can also be considered as another successful application of the four-point probe method, extending it to nanomaterials science to study substrate-supported nanomaterials.

To clarify this point, we have added the following text on page 3 and 4 in the revised manuscript.

- “The method that is currently most widely used to extract nanomaterial information from measurements obtained for substrate-supported nanomaterial samples can be summarized

as a two-point probe method, in which traditional data-processing techniques, such as spectrum subtraction and ratioing, are applied to two interrelated spectra measured for a covering nanomaterial and bare substrate to highlight the spectral features related to the nanomaterial. However, using a two-point probe method, the influence from substrate signals can only be weakened rather than completely removed, so the information obtained about a nanomaterial is not quantitative. There is a growing consensus that it is impossible to extract pure nanomaterial information from just two spectra using simple algebra without any prior knowledge about distinguishing spectral features. Because the two-point probe method cannot provide quantitative information about a nanomaterial, it seems that a method with more probe points, like the four-point probe method, may overcome this limitation. In fact, the feasibility of this logic has been demonstrated in various fields; for instance, the four-point probe method has been successfully implemented in materials science to precisely determine the electrical resistance of solid-state matter by excluding contributions from parasitic contact resistances [5], and also in radio astronomy as the chop-nod method [6] to detect faint astronomical sources by ground-based telescopes despite the bright, variable sky background. Learning from these successful examples, here we propose the virtual substrate method, which is an extension of the four-point probe method to nanomaterials science, to study substrate-supported nanomaterials without influence from substrate signals. Using the virtual substrate method, the equivalent transmission configuration experiment can be realized from a combination of four interrelated measurements in reflection configuration.”

Comment 6: *The explanation of the chop-nod method on page 3 needs significant improvement.*

Response to Comment 6: We have rewritten this part on page 3 as shown above (see **Response to Comment 5**).

Comment 7: *Page 3, line 50, the text erroneously states that the signal-to-noise is improved. The random noise cannot be reduced by the procedure, indeed, by the subtraction it is increased here by approximately by 1.4. The background may be removed increasing the signal-to-background ratio. Maybe this is what the authors were thinking. The text requires correction.*

Response to Comment 7: We apologize for the inappropriate phrase “signal-to-noise” in the previous manuscript. We have replaced this term in the revised manuscript with “signal-to-background”.

In accordance with this comment, we have added the following text on page 17 in the revised manuscript.

- “..., the virtual substrate method outrivals conventional methods based on core-level signals in **signal-to-background** ratio by orders of magnitude,...”

Comment 8: Page 3 line 53 the word “which” refers back to the word “signals”. This is clearly nonsense. What the authors may refer to is the dimensional size being 20 orders of magnitude smaller. This clause should be deleted as irrelevant.

Response to Comment 8: We apologize for this oversight. We have deleted the corresponding sentence and rewritten this part of the manuscript as described above (see **Response to Comment 5**).

Comment 9: The ordinate labelled “elastic transmission” in Figure 2 is unclear. Either here or in the caption, the algebraic symbol defined in the text should be given – i.e. how this relates to $J_{\Delta A}$, etc.

Response to Comment 9: We apologize for the unclear labels in Figure 3 (Figure 2 in the original manuscript). We have redrawn Figure 3 and edited its caption in the revised manuscript as follows.

Figure 3 | Elastic transmission of mono- and bilayer graphene. Elastic transmission ($T_{\Delta N/\Delta S}(E)$, that is $J_{\Delta N}(E)/J_{\Delta S}(E)$) of monolayer graphene and bilayer graphene measured by the virtual substrate method for primary electron energies of 10 and 15 keV. The elastic transmission was calculated by the extended Mermin (EM), Monte Carlo (MC) and time-dependent density functional theory (TDDFT) methods. The total transmission [23,24] was measured by low-energy electron point-source (LEEPS) microscopy corrected to the elastic transmission by the MC method. The elastic transmission was measured by the overlayer method for graphene/gold at 2.3 eV, associated with gold

surface plasmons (Overlayer-SP), and for graphene/SiO₂ samples at 78 and 503 eV (Overlayer).

Comment 10: *In the method on page 18, it is noted that similar regions are used on both the bare and graphene-covered substrates. How does the user know when the chosen regions are suitably similar. What defines this, both the experimental choice of the grain in the instrument when conducting the measurements and in terms of the number of degrees of crystal mis-orientation involved.*

Response to Comment 10: The identity of the gold grains with and without covering graphene is extremely important to perform the virtual substrate measurement (nano-chop-nod measurement in the original manuscript) with high precision. An electron backscatter diffraction study showed that the uncertainty range of relative orientations for one type of gold grains with and without a covering graphene layer is about 0.5°, while the relative orientation between two types of gold grains used in the present virtual substrate measurements is about 4°. In fact, besides the relative orientations of the gold grains, other differences between gold substrates such as variations in crystal quality and surface morphology will also lower the accuracy of transmitted spectra. Therefore, the easiest way to consider all these contributions together is to verify whether or not the gold grains considered as the same type can produce the same reflected SE spectra (white electrons). In this work, the reliability of these selected incident positions was verified by the consistency of reflected SE spectra measured at these positions before transferring the graphene sheets onto half of them.

To clarify this point, we have added the following text on page 12 in the revised manuscript.

- “First, four raw spectra are measured by selecting incident positions on different crystallographic orientations in the bare substrate and similar regions covered by graphene sheets. The reliability of these selected incident positions was verified by the consistency of the raw spectra measured at these positions before transferring the target nanomaterial sheets onto half of them. Atomic force microscopy (AFM) was used to confirm the absence of wrinkles in the covering nanomaterial layer.”

Responses to the second Referee's comments

Comment 1: *I really like the concepts behind this work, but found a number of shortcomings. The first is simply that, in its present form, the work doesn't excite one as much as it could. A second issue (perhaps the cause of the first) is that some of the concepts are not clear in the main text. It is only in the Methods section that it is shown why one samples the substrate and substrate+nanomaterial for at least two different substrate types (4 spectra), i.e., this removes the reflectivity difference between substrate and nanomaterial. Figure 1 is quite complex, but never really makes this point.*

I don't think that this is a Nature Communications in its present form. However, I like the idea and it would be valuable to have it disseminated widely. Perhaps this can be reconsidered after extensive edits to clarify the concepts. Some of this might involve moving material from Methods to the main text (and a large portion of Methods probably should move to Supplemental).

Response to Comment 1: We apologize for our unclear description of the concept of the method in the original manuscript. We have reconstructed the manuscript and added two new parts in the results section entitled “**The virtual substrate method; a new type of four-point probe method**” and “**White electrons in the virtual substrate method**” to clarify the concepts of the method. We have also added a new figure (Figure 1) to intuitively demonstrate how the reflectivity difference between substrate and nanomaterial was removed using the four-point probe measurement. In the revised manuscript, we use the term “four-point probe method” to describe the concept behind this work; i.e., the general idea of basic acquisition of spectra with and without X that are assumed to be linear additions of effects to allow removal by the use of simple algebra. In this case, the virtual substrate method (nano-chop-nod method in the original manuscript) can be considered as an extension of the general four-point probe concept to nanomaterials science to remove substrate contributions. The virtual substrate method is analogous to the four-point probe resistance measurement technique in materials science used to remove parasitic contact resistances and the chop-nod method used in radio astronomy to remove thermal background.

To highlight the concept behind this work (the four-point probe method), we have added a new part in the results section entitled “The virtual substrate method; a new type of four-point probe technique” on pages 5–9.

- “**The virtual substrate method; a new type of four-point probe technique.** Although the virtual substrate method is not restricted to surface analysis, the implementation of this principle shown in Fig. 1 is based on surface electron spectroscopy techniques. The raw spectra represent the evolution of a primary electron beam inside a sample driven by the interaction of the sample with moving electrons. From the viewpoint of mathematics, the energy spectrum $J_0(E)$ of a normally incident electron beam can be described by a special vector with one non-zero element

representing the incident electron energy. First, a measurement of a bare substrate (substrate A) is considered, where the substrate acts as the scatterer that emits the reflected electrons and SEs. Such a process is essentially a modification of $J_0(E)$, transforming the monochromatic incident electrons into the emitted “white electrons”. Therefore, the scattering process can be described by the matrix \mathbf{R} , and the reflected spectrum from the substrate can be written as $J_{S(A)}(E) = \mathbf{R}J_0(E)$.

Next, we consider the case where a nanomaterial is placed on the top of the substrate, which is also the configuration used for conventional reflection spectroscopy. The electron beam is first incident on the nanomaterial and produces SEs and partially reflected electrons, which can be described by the material-dependent matrix \mathbf{R}^N , so the reflection spectrum can be denoted as $\mathbf{R}^N J_0(E)$. In addition to this reflection process, a transmission process also occurs, which is denoted by the material-dependent matrix \mathbf{T}^N . These transmitted electrons with spectrum $\mathbf{T}^N J_0(E)$ then interact with the underlying substrate and lead to the reflected spectrum $\mathbf{R}^N \mathbf{T}^N J_0(E)$. These substrate-reflected electrons subsequently pass through the nanomaterial on the top of the substrate, creating the new spectrum $\mathbf{T}^N \mathbf{R}^N \mathbf{T}^N J_0(E)$. In this work, we only consider the approximation to the first order; i.e., we neglect any further reflection between the nanomaterial and substrate. Furthermore, we can approximate \mathbf{T}^N as unity for high-energy incident electrons (the first \mathbf{T}^N starting from the right in $\mathbf{T}^N \mathbf{R}^N \mathbf{T}^N J_0(E)$), which physically corresponds to the complete transmission of high-energy electrons through the ultra-thin nanomaterial. Therefore, the measured spectrum for a nanomaterial on a substrate can be written as

$$J_{N(A)}(E) = \mathbf{R}^N J_0(E) + \mathbf{T}^N \mathbf{R} J_0(E). \quad (1)$$

Physically, this means that in a single measurement (i.e., conventional reflection measurement), the obtained spectrum for the substrate-supported nanomaterials include contributions from several sources: (i) $\mathbf{R}^N J_0(E)$, the reflection from the nanomaterial and typically SEs originating from the interaction of the high-energy monochromatic incident electrons and the nanomaterial; and (ii) $\mathbf{T}^N \mathbf{R} J_0(E)$, the transmitted spectrum originating from the substrate-reflected electrons $\mathbf{R} J_0(E)$. This greatly complicates the data processing and prevents extraction of the full information of the target nanomaterials. In a traditional two-point probe measurement (represented by the processes inside the red dashed box in Fig. 1a), measurements were performed on both the substrate and substrate-supported nanomaterial. For the reasons discussed above, the substrate reflection is measured separately and the second term in equation (1) can be written as $\mathbf{T}^N J_{S(A)}(E)$. Thus, we will have

$$J_{N(A)}(E) = \mathbf{R}^N J_0(E) + \mathbf{T}^N J_{S(A)}(E), \quad (2)$$

where $J_{N(A)}(E)$ and $J_{S(A)}(E)$ are the measured spectra for the substrate-support nanomaterial and bare substrate, respectively. \mathbf{R}^N and \mathbf{T}^N are the reflection and transmission matrices for the nanomaterial, respectively. These matrix elements are quantitatively linked to the e–e interaction. Therefore, by solving \mathbf{R}^N and \mathbf{T}^N using linear equations, we can obtain complete information about the target nanomaterial.

However, even neglecting all off-diagonal elements of \mathbf{T}^N (i.e., for an ultrathin nanomaterial such as mono- or bilayer graphene), the number of unknown variables, that is, $r_{i,j}$ ($j = j_0$) for matrix elements of \mathbf{R}^N in a given column referring to primary incident electron beam energy and $t_{i,j}$ ($i = j$) for matrix elements of \mathbf{T}^N on the principal diagonal, is twice the number of equations in equation (2). That is, there is only one equation with two unknowns at a given energy. Therefore, to obtain a solution, we need an additional set of measurements, which can be obtained by collecting another set of measurements using a different substrate. As shown in Fig. 1a, we then performed the traditional two-point probe measurement with an additional substrate (substrate B) to obtain another system of linear equations, such that

$$J_{N(B)}(E) = \mathbf{R}^N J_0(E) + \mathbf{T}^N J_{S(B)}(E). \quad (3)$$

With the number of variables now equal to the number of equations (equation (2) and (3)), we can solve the matrices \mathbf{R}^N and \mathbf{T}^N and thereby obtain complete information for a target nanomaterial.

Combining equation (2) and (3) determined according to the four-point probe method, we will have

$$J_{\Delta N}(E) = \mathbf{T}^N J_{\Delta S}(E), \quad (4)$$

where $J_{\Delta N}(E)$ and $J_{\Delta S}(E)$ are the difference spectra, which can be obtained by subtracting two measured spectra for a substrate-supported nanomaterial ($J_{N(A)}(E)$ and $J_{N(B)}(E)$), and two spectra for the substrates ($J_{S(A)}(E)$ and $J_{S(B)}(E)$). The mathematical expression of $J_{\Delta N}(E)$ is $\mathbf{T}^N(\mathbf{R}_A - \mathbf{R}_B)J_0(E)$, while that of $J_{\Delta S}(E)$ is $(\mathbf{R}_A - \mathbf{R}_B)J_0(E)$, where \mathbf{R}_A and \mathbf{R}_B are matrix descriptions of the reflection process of substrate A and B, respectively, and can be further simplified as $\mathbf{T}^N \delta \mathbf{R} J_0(E)$ and $\delta \mathbf{R} J_0(E)$, respectively, where $\delta \mathbf{R}$ represents a “virtual substrate” whose contribution is equivalent to the responses of two substrates to injected electrons ($\delta \mathbf{R} = \mathbf{R}_A - \mathbf{R}_B$). It is obvious that $J_{\Delta S}(E)$ and $J_{\Delta N}(E)$ are the responses of the bare substrate system and nanomaterial/substrate system, respectively, to the virtual substrate, which are not related to the concrete substrate used in these systems. The physical meaning of the two difference spectra $J_{\Delta S}(E)$ and $J_{\Delta N}(E)$ then becomes apparent; they are the initial and final states, respectively, for white electrons with the expression $\delta \mathbf{R} J_0(E)$ travelling through a nanomaterial. Therefore, the ratio $J_{\Delta N}(E)/J_{\Delta S}(E)$ (that is, $\mathbf{T}^N \delta \mathbf{R} J_0(E) / \delta \mathbf{R} J_0(E)$) directly reveals the quantitative e-e interaction information for matrix \mathbf{T}^N . \mathbf{T}^N is typically a lower triangular matrix that can simply be split into two matrices:

$$\mathbf{T}^N = \begin{pmatrix} t_{1,1} & & & \\ t_{2,1} & t_{2,2} & & \\ \cdots & \cdots & \ddots & \\ t_{n,1} & t_{n,2} & \cdots & t_{n,n} \end{pmatrix} = \begin{pmatrix} t_{1,1} & & & \\ & t_{2,2} & & \\ & & \ddots & \\ & & & t_{n,n} \end{pmatrix} + \begin{pmatrix} 0 & & & \\ t_{2,1} & 0 & & \\ \cdots & \cdots & \ddots & \\ t_{n,1} & \cdots & t_{n,n-1} & 0 \end{pmatrix}. \quad (5)$$

The first matrix, which includes only the elements on the principal diagonal, reflects the elastic electron transmission information of the nanomaterial. Elements $t_{i,j}$ ($i = j$) are the elastic electron transmission of the nanomaterial, which converge to 1 as the index

(electron energy) increases. The second matrix includes only those elements $t_{i,j}$ ($i > j$) below the principal diagonal and is a sparse matrix whose non-zero entries are located in two major regions. One major region is confined to a diagonal band below the main diagonal, providing information about inelastic scattering processes. The lower bandwidth of this region depends on the electron energy and nanomaterial thickness. The other major region is located far from the main diagonal near the bottom left corner of the matrix and describes the production of SEs in inelastic scattering processes. The intensities of SEs directly reflect the energy loss behaviour of the nanomaterial. For an ultra-thin target nanomaterial, such as mono- or bilayer graphene, the \mathbf{T}^N matrix mainly contains contributions from the first matrix term and can be treated as the elastic electron transmittance of the target nanomaterial. For a thick target nanomaterial, such as few-layer graphene ($n > 5$), the second matrix term is dominant, reflects the accompanying SE emission at low energy and provides energy loss information about the target nanomaterial.

In fact, there is another more intuitive way to demonstrate the principle of the virtual substrate method. As shown in Fig. 1b, the relationship between the measured spectra ($J_{S(A)}$, $J_{N(A)}$, $J_{S(B)}$ and $J_{N(B)}$) and the determined elements in the matrices \mathbf{T}^N and \mathbf{R}^N can be visualized as a finite number of lines that pass through the two points ($J_{S(A)}$, $J_{N(A)}$) and ($J_{S(B)}$, $J_{N(B)}$). These intersection points in the lines correspond to every energy in the measured spectra, whose slope and intercept are \mathbf{T}^N and $\mathbf{R}^N J_0(E)$, respectively. According to this relationship, when the intensities of four interrelated raw spectra in the form of the two points ($J_{S(A)}$, $J_{N(A)}$) and ($J_{S(B)}$, $J_{N(B)}$) are considered inputs, then the slope and intercept of the determined lines, \mathbf{T}^N and $\mathbf{R}^N J_0(E)$, respectively, are the outputs, and include only the properties of the nanomaterial. A more intuitive description is that the virtual substrate method converts a line determined from the absolute intensities of four interrelated raw spectra at a given energy in measurable space into a point in parameter space (slope–intercept parameterization for a straight line), where the slope (i.e., diagonal elements of \mathbf{T}^N) and intercept (i.e., $\mathbf{R}^N J_0(E)$) can be considered the equivalent transmitted electron spectrum and equivalent reflected electron spectrum for a free-standing target nanomaterial, respectively.”

To illustrate these concepts, we have added the following Figure 1 to the manuscript together with its caption.

Figure 1 | Visualization of the virtual substrate method. **a**, Schematic diagram of the virtual substrate method implemented in surface analysis. A combination of four interrelated spectra measured for two slightly different bare substrates ($J_{S(A)}$, $J_{S(B)}$) and a target nanomaterial supported on these two substrates ($J_{N(A)}$, $J_{N(B)}$) is used in the virtual substrate method. Different groups defined as “substrate” and “nanomaterial” (surrounded by black boxes) are classified from the spectra measured for the bare substrate or nanomaterial. A traditional two-point probe measurement is indicated by a red dashed box. **b**, Visual representation of the virtual substrate method using a 3D coordinate system, where two spectra obtained using a traditional two-point probe measurement are plotted in pairs orthogonally along the x - and y -axes and share one electron energy axis (z -axis). Two spectra measured for bare substrates (blue and green dots) are plotted in the x - z plane (substrate plane), and the other two spectra measured for the nanomaterial supported by substrates (blue and green dots) are plotted in the y - z plane (nanomaterial plane). According to the virtual substrate method, the covering nanomaterial information is included in the lines that pass through two points whose x and y coordinates are the intensities of the two spectra in the traditional two-point probe measurements for different substrates at a given energy. The intercept of these lines, $\mathbf{R}^N J_0(E)$, is plotted in the y - z plane (nanomaterial plane) as red dots. One line (purple) at energies E_i is presented together in the x - z and y - z planes (thin purple lines) along with the deviations in these shallow lines (thick cyan line). At E_i , two known points ($J_{S(A)}$, $J_{N(A)}$) and ($J_{S(B)}$, $J_{N(B)}$) obtained by traditional two-point probe measurements with different substrates and the intercept point $(0, \mathbf{R}^N J_0)$ are presented as large black dots.

Comment 2: L41: *There is always a surface contribution because surface/volume ratio is large in nanomaterials. It only becomes apparent much later (in Methods), that it's the surface reflection that is the issue.*

Response to Comment 2: We apologize for our misguided description of the “surface properties” of the nanomaterial. We have rewritten this part on line 43 on page 3 as follows.

“..., because doing this causes the obtained information to be related to **the properties of the surface atomic layer of the nanomaterial**, rather than the overall properties of the entire nanomaterial. **Generally, the overall properties of the entire nanomaterial can only be measured by techniques using transmission configuration.**”

Comment 3: *Fig. 2: Are the LEEPS results obtained by the authors or from the literature? The way the caption is written and referenced makes it sound as though the authors did the measurements; however, this is a fairly specialized measurement, so it seems likely that the result is from the literature. If these are literature results, the figure label should clearly state that the results are from Refs. 23,24.*

Response to Comment 3: We apologize for omitting a citation for the LEEPS results. We have redrawn Figure 3 (Figure 2 in the original manuscript) and corrected the citation in the corresponding caption (see **Response to Comment 9** for the first referee).

Comment 4: *Fig. 3: Same issue as Fig. 2, with regard to VLEED and LEEM results. It is essential to make clear whether you performed the measurement based on the techniques described in the reference, or took the actual results from the reference.*

Response to Comment 4: We apologize for omitting citations of previous work. We have redrawn Figure 4 (Figure 3 in the original manuscript) and included the relevant citations in the corresponding caption.

Figure 4 | Material characterization using the virtual substrate method. a, Top: transmitted spectra for one-, four-, six-, eleven- and fourteen-layer graphene for primary electron energies of 10 keV (dashed lines) and 15 keV (solid lines). Insets show the transmitted spectra at 15 keV without an offset. Bottom: Electron transmission spectra (T_{VLEED}) [25] measured by very-low-energy electron diffraction (VLEED) and transmission data (T_{LEEM}) [26] estimated from the reflectivity spectra ($T_{\text{LEEM}} = 1 - R_{\text{LEEM}}$) obtained from low-energy electron microscopy (LEEM)...

Comment 5: L131: I couldn't figure out the phrase "biased estimated thickness"; what biases the estimate?

Response to Comment 5: We apologize for the unclear phrase "biased estimated thickness" in the original manuscript. We have replaced it with "mainly provided by the errors introduced by using the dielectric function of the well-known jellium model to describe the electrical properties of single-crystal graphene." on page 13.

Responses to the third Referee's comments

Comment 1: *The method is however also quite restricted to specific conditions. For example one needs two different substrates in order to obtain a difference spectrum. However growing the same nanomaterial on different substrates would in most case result in films of different structures as well as different thickness. Identical films on different substrate like graphene on Au is not generally available, which limits the applicability of this method.*

Response to Comment 1: It is true that identical films on different substrates are not generally available. Naturally, the difference between films on different substrates will lower the accuracy of transmitted spectra obtained by the virtual substrate method (call the nano-chop-nod method in the original manuscript). Therefore, for quantitative study of a target nanomaterial, it is best to transfer the target nanomaterial onto the tailor-made substrate, which is polycrystalline gold in this work. For samples with a given substrate, usually only non-quantitative information about the target nanomaterial can be obtained.

To clarify this point, we have added the following text on page 16 and 17 in the discussion section of the revised manuscript.

- “The experimental conditions required for a virtual substrate measurement are similar to those of traditional measurements except for the substrate on which the target nanomaterial is supported. A tailor-made substrate is important in the virtual substrate method; in this work, the substrate is polycrystalline gold. In fact, strict requirements for the substrate are only necessary for highly precise, high-speed quantitative studies of target nanomaterials. To obtain low-precision measurements, virtual substrate measurements can be performed using almost any two substrates with different element composition, surface morphology, and crystal quality. However, polycrystalline metal substrates are the best choice for highly precise, high-speed quantitative studies of target nanomaterials because of their similar surface barrier and degree of interaction with the covering nanomaterial. For instance, a hole-patterned SiO₂ substrate has also been used to investigate multilayer graphene by virtual substrate measurements. Four spectra measured for flat and hole regions of the SiO₂ substrate without and with a covering graphene layer are used in this measurement instead of spectra measured for two different types of gold grains. The results of this measurement were broadly in line with those obtained for a graphene/polycrystalline gold system, but had very low precision. This demonstrates that we are able to design other tailor-made substrates in accordance with specific conditions; however, tailor-made substrates that can produce nearly identical reflected SE spectra at two determined measurement points are necessary to perform virtual substrate measurement.

For research purposes, the target nanomaterial can be transferred onto the tailor-made substrate; however, for industrial purposes, it is usual to study a target nanomaterial on a given substrate (any arbitrary substrate). Stringent demands for the substrate are essential

to extract pure information of target nanomaterials with high precision through virtual substrate measurements. These stringent demands for the substrate are bound to restrict the scope of this technique in wider applications, especially industrial ones. However, there are exceptions even in industry; for instance, the virtual substrate method will be a very efficient tool to quantitatively investigate passive films on stainless steel, which is a typical nanomaterial/polycrystalline substrate system.”

Reviewers' comments:

Reviewer #1 (Remarks to the Author):

Virtual substrate method for nanomaterials characterization

by Bo Da, Jiangwei Liu, Mahito Yamamoto, Yoshihiro Ueda, Kazuyuki Watanabe, et al.

The authors have made extensive revisions to answer the issues raised. However, the new text: "Although multi-point probe measurements have been used to obtain information from measured spectra for many years already, like multi-spectral approaches using Auger electron microscopy [30], regardless of how many interrelated spectra are involved, the concept behind the virtual substrate measurement presented in this work is quite different from the standard data treatment methods represented by subtraction and ratioing techniques. Firstly, the essence of standard data treatment is to screen out useful data points from measured spectra, and draw a conclusion according to the information obtained from these selected data points. In contrast, the essence of the virtual substrate method is to make "useless" data points become useful. That is, with the help of the virtual substrate method, data points that are meaningless alone in individual measurements can be converted into useful data points with direct physical meaning. This breaks out from the old pattern of thinking that useless and useful data points are completely isolated from each other. Secondly, creating trackable white electrons from the interrelation between absolute intensities of interrelated measured spectra instead of expecting a realistic signal in a measured spectrum (relative intensities in one spectrum) is a completely new approach in surface analysis that helps data treatment to move away from depending on identifying signal features with the naked eye." is, in answer to Comment 3, unhelpfully self-inflatory and all this text should be replaced by:

"Although multi-point probe measurements have been used to obtain information from measured spectra for many years already, like multi-spectral approaches using Auger electron microscopy [30], the concept behind the virtual substrate measurement presented in this work is a new development."

In response to Comment 10, the authors have said what they did in the text of their response but then in the actual revision to the text of the paper appear to explain what "similar" means not by that explanation but by using the words "similar" and "consistent", i.e. they have replaced fog by more of the same fog – one may ask now, how consistent is consistent? The authors need to explain to the reader what the reader should do to obtain suitable results as in the text to the reviewer, i.e the relative orientations should be better than 4 degrees, etc, etc.

This work requires some further revision.

Reviewer #2 (Remarks to the Author):

The manuscript is much clearer after revisions, and the title is more appropriate. It's valuable work. However, the techniques seem not as generally applicable as the abstract says, and do involve quite a bit of input from auxiliary calculations. In the end, unless the generality can be demonstrated, I think that this is a topic best suited for a more focused journal than Nature Communications.

There are still some issues of grammar and clarity:

Main text:

In the description of the method, present tense throughout would be most appropriate (in my

opinion). Therefore:

L127: "measurements were" should be "measurements are"

L143: "performed" should be "perform"

L191-196: The "intuitive description" is valid only for thin layers, where it is appropriate to neglect the inelastic contribution. Perhaps this caveat should be included.

L208-220: This section is not very clear. The terminology "approximate separation of energy channels" is unclear to a broad audience (me), as is "trackable white electrons" (the first phrase is clarified in the supplementary material). I've understood the white electron concept, but not from these lines.

L290: "showed" should be "show." Also, the sentence (L288-291) still makes it sound as though the authors have performed the point source microscopy, at least until one looks at the references. A rearrangement might be clearer: "The virtual substrate measurements show excellent agreement with existing electron point source microscopy at 66 eV [23] and 100 eV [24] and with elastic transmission measurements performed in this work using the AES overlayer method [21] at 2.3 eV (gold surface plasmon), 78 eV (Si LVV), and 503 eV (O KLL) [22]."

L328 (and surrounding discussion): Does "useless" signify the normal background signal? Are the "useful" data points considered to be those under a peak of limited extent in energy? This section could be written more clearly.

L332: "trackable white electrons" is still a mysterious phrase to me.

L500-501: I would suggest placing reference [23,24] after the word "microscopy," and inserting a reference to supplementary material after "method" (unless the LEEPS authors were the ones who used the MC method).

Supplementary Material:

L66: keV should be eV

L71: CAM should be CMA

Fig. 7: Flow chart top box should read "virtual crystal measurement"

Reviewer #3 (Remarks to the Author):

The authors have carefully addressed my comment. I have no further suggestions.

We appreciate the constructive feedback from the editor and referees. We have individually addressed all of the comments and revised the manuscript accordingly. In this document, the referees' comments are shown in italics, and the responses in normal typeface. In the revised manuscript and Supplementary Information, changes are shown in red.

Responses to the first referee's comments

Comment 1: *The authors have made extensive revisions to answer the issues raised. However, the new text; “Although multi-point probe measurements have been used to obtain information from measured spectra for many years already, like multi-spectral approaches using Auger electron microscopy [30], regardless of how many interrelated spectra are involved, the concept behind the virtual substrate measurement presented in this work is quite different from the standard data treatment methods represented by subtraction and ratioing techniques. Firstly, the essence of standard data treatment is to screen out useful data points from measured spectra, and draw a conclusion according to the information obtained from these selected data points. In contrast, the essence of the virtual substrate method is to make; useless; data points become useful. That is, with the help of the virtual substrate method, data points that are meaningless alone in individual measurements can be converted into useful data points with direct physical meaning. This breaks out from the old pattern of thinking that useless and useful data points are completely isolated from each other. Secondly, creating trackable white electrons from the interrelation between absolute intensities of interrelated measured spectra instead of expecting a realistic signal in a measured spectrum (relative intensities in one spectrum) is a completely new approach in surface analysis that helps data treatment to move away from depending on identifying signal features with the naked eye;” is, in answer to Comment 3, unhelpfully self-inflatory and all this text should be replaced by: “Although multi-point probe measurements have been used to obtain information from measured spectra for many years already, like multi-spectral approaches using Auger electron microscopy [30], the concept behind the virtual substrate measurement presented in this work is a new development”*

Response to Comment 1: We agree that the previous description is unhelpful. We have rewritten this paragraph according to referee's suggestion, as follows “Although multi-point probe measurements have been used to obtain information from measured spectra for many years already, like multi-spectral approaches using Auger electron microscopy [31], **the concept behind the virtual substrate measurement presented in this work is a new development.**”.

Comment 2: *In response to Comment 10, the authors have said what they did in the text of their response but then in the actual revision to the text of the paper appear to explain what “similar”; means not by that explanation but by using the words “similar”; and “consistent”;, i.e. they have replaced fog by more of the same fog; one may ask now, how consistent is consistent? The authors need to explain to the reader what the reader should do to obtain suitable results as in the text to the reviewer, i.e the relative orientations should be better than 4 degrees, etc, etc.*

Response to Comment 2: We apologize for this unclear description. We have rewritten this paragraph on page 12,

- “Generally, the uncertainty range of relative orientations for one type of metal grains with and without a covering graphene layer should be within 0.5° , while the relative orientation between two types of metal grains should be larger than 4° . The reliability of these selected incident positions was verified by the consistency of the raw spectra measured at these positions before transferring the target nanomaterial sheets onto half of them, and the relative errors should be within 5%. Atomic force microscopy (AFM) was used to confirm the absence of wrinkles in the covering nanomaterial layer. The criteria for selecting these measurements points are presented in Supplementary information.”

Furthermore, a custom-made software have been developed by us to select measurement points semi-automatically in the virtual substrate method. The principle part of the handbook of this custom-made software have been added in Supplementary Information as “Supplementary Note2. The criteria for selecting measurements points when using polycrystalline gold substrate in virtual substrate measurement.” with five new figures (Supplementary Fig. 9-13).

Responses to the second Referee's comments

Comment 1: *The manuscript is much clearer after revisions, and the title is more appropriate. It's valuable work. However, the techniques seem not as generally applicable as the abstract says, and do involve quite a bit of input from auxiliary calculations. In the end, unless the generality can be demonstrated, I think that this is a topic best suited for a more focused journal than Nature Communications.*

Response to Comment 1: We agree that this new method requires a fair amount of auxiliary work. However, auxiliary calculations (auxiliary calculations represents those Monte Carlo simulations used to remove ISP and SEE contributions) are only necessary for highly precise quantitative studies of target nanomaterial, like measuring elastic electron transmission of mono- and bilayer graphene as presented in this work. For qualitative purpose, auxiliary calculations does not play a decisive role. Without auxiliary calculations, in most cases, we can still do nanomaterial characterization.

To demonstrate the generality and applicability of the virtual substrate method, efforts in three aspects were made.

Firstly, another example of using virtual substrate method to distinguish substrate-supported mono- and bilayer graphene by electron-beam technique have been added in the Discussion section on pages 16.

- “In fact, besides the accompanying SE features in transmitted spectra, the electronic properties of a target nanomaterial determined by a virtual substrate measurement can also be used as a descriptor for nanomaterial characterization. To demonstrate this, virtual substrate measurements were performed for investigating mono- and bilayer molybdenum disulfide (MoS_2) samples, which have different electronic properties in the low energy range as identified by optical spectroscopy [32]. The determined transmitted spectra for mono- and bilayer MoS_2 are presented in Fig. 5. Consistent fluctuation of electron energy caused by a combination of inelastic scattering, accompanying SE and the diffraction effect is observed in the transmitted spectra for mono- and bilayer MoS_2 over the whole energy range, except for between 2 eV and 5 eV, as highlighted by blue arrows. When the electron energy exceeds 2 eV, the intensity of transmitted spectra for monolayer MoS_2 decreases gradually with increasing electron energy until the energy reaches about 5 eV. For bilayer MoS_2 , the intensity of transmitted spectra decline sharply until this behavior suddenly stops at about 2.5 eV. This variation between mono- and bilayer MoS_2 may be caused by different e-e interactions near the band gap energy, different out-of-plane properties, or different interfacial properties when mono- and bilayer MoS_2 contact with the gold substrate (these mechanisms will be discussed elsewhere). Such variation in measured transmitted spectra can be used as an indicator in electron-beam techniques to distinguish substrate-supported mono- and bilayer MoS_2 .

Furthermore, the differences between mono- and bilayer MoS₂ are more obvious when organizing these transmitted spectra in the form of IMFP by reversing equation (1); i.e., $\lambda_{\text{IMFP}} = -(nd_0)/(\ln T_n \cdot \cos\theta)$. The determined IMFPs for mono- and bilayer MoS₂ agreed well over the whole energy range, except for between 2 eV and 5 eV, which is consistent with the observed transmitted spectra.”

Accordingly, we have added the following Figure 5 to the manuscript together with its caption.

Figure 5 | Electronic properties of monolayer and bilayer MoS₂. Top: Transmitted spectra for mono- and bilayer MoS₂ obtained at a primary electron energy of 20 keV. Two different MoS₂ sheets supported on the same batch of substrates, denoted as S1 and S2, were used in virtual substrate measurements with energy ranges of up to 12 eV in 0.1 eV increments and 120 eV in 0.5 eV increments, respectively. Bottom: Corresponding IMFP, λ_{IMFP} , of mono- and bilayer MoS₂ for a primary electron energy of 20 keV.

Secondly, a custom-made software for selecting measurement points have been developed. How to select measurement points on a given substrate, for instance, polycrystalline gold substrate, is fundamentally important to be able to implement the virtual substrate method with high precision, and also, to an extent, determine whether or not the virtual substrate measurement requires certain skills, which directly affects the generality of this approach.

In order to solve this question, we successfully developed a custom-made software to help users semi-automatically, quickly and easily select these measurement points according to the intensities of substrate grains in a SEM image. The principle part of the handbook of this custom-made software have been added in Supplementary Information as “**Supplementary Note2. The criteria for selecting measurements points when using polycrystalline gold substrate in virtual substrate measurement.**” with five new figures (Supplementary Fig. 9-13).

Thirdly, another way to implement the virtual substrate method to study a nanomaterial sample with a given substrate have been introduced in the Discussion section. Using this new approach, target nanomaterial on a given substrate as long as it displays fairly stable intensity distributions in a SEM image can be investigated from monochromatic SEM image. The corresponding description is presented on page 18.

- “In fact, there is another way to implement this virtual substrate method by which almost any given nanomaterial/substrate combination with a substrate that is not completely uniform, like a single crystal, can be investigated just from a “monochromatic” SEM image without any designed substrate, and even without selecting measurement points. In this spectral imaging approach, the pixels of a SEM image formed by detected electrons at a given energy are considered as pixel sized “fictitious grains” and used to perform the virtual substrate measurement as described in the Supplementary Information. This new approach has allows us to study a target nanomaterial on a given substrate as long as it displays fairly stable intensity distributions in a SEM image. A limitation of this approach is the huge amount of data generated during measurement; however, it does represent a possible future direction for the virtual substrate method.”

Besides that, the principle of this new implementation is also presented in Supplementary Information on page 29 as follow:

Supplementary Figure 14. Virtual substrate measurement without selecting measurement points. **a**, SEM image obtained from 100 ± 2 eV electrons. The bare gold substrate and graphene-covered regions are indicated by red and blue dashed lines, respectively. **b**, The proportion of pixels over 30 levels (relative intensity on the bottom x -axis) according to their intensities (electron count on the top x -axis) on the bare gold substrate and monolayer graphene-covered regions. The fictitious measurement points are

marked as pink diagonal, yellow, red diagonal and blue crosses representing S(A), S(B), N(A) and N(B), respectively. **c**, The accumulated proportion of pixels on the bare gold substrate and graphene-covered regions plotted according to their intensities. The lines of best fit for the data points with an accumulated proportion value in the range of 15%–85% for the bare gold substrate and graphene-covered regions are presented. The deviation of intensities of the calculated accumulated proportions of 80% and 20% for the bare gold substrate and graphene-covered regions are indicated by thick black lines.

Supplementary Note3. Performing virtual substrate measurements by scanning SEM image

The reason why we need to select measurement points before performing a virtual substrate measurement is to verify the identity of reflected SE spectra produced at two determined measurement points like S(A) and N(A). Take a polycrystalline gold substrate for instance, as shown in Supplementary Fig. 14.

It is well known that a covering graphene layer does not affect the relative order of the intensities of gold grains. Therefore, two gold grains that share the same ranking as bare and graphene-covered gold substrates have the potential to produce the same reflected SE spectra after removing the covering graphene layer. Breaking our old pattern of thinking, the pixels of a SEM image formed by detected electrons at a given energy (Supplementary Fig. 14a) can be considered as pixel-sized “fictitious grains”. It should be noted that we do not expect that the intensities of these pixel-sized “fictitious grains” represent the behavior of whole spectra at all energies, which is expected when using realistic gold grains. In this case, the pixel-sized “fictitious grains” that share the same ranking of intensity distribution of pixels in the bare gold substrate and graphene-covered regions can be used to perform virtual substrate measurements only at this given energy instead of selecting measurement points on realistic gold grains. The intensity distribution of pixels in the bare gold substrate and graphene-covering regions normalized by the total number of involved pixels are plotted in Supplementary Fig. 14b. The distribution of these pixel-sized “fictitious grains” in the bare gold substrate region is broader than that in the graphene-covered region because of the attenuation effect of the covering graphene layer. Likewise, the presence of a graphene layer does not change the ranking of pixel-sized “fictitious grains”. The pixel-sized “fictitious grains” that have the same ranking in the bare gold substrate and graphene-covered regions can be considered as the same pixel-sized “substrate”, analogous to selecting the same type of gold grains. For instance, the pixel-sized “fictitious grains” with rankings of 20% and 80% can be considered as two different substrates, allowing virtual substrate measurements to be performed. In this case, the difference spectra (only one data point at 100 eV) can be written as $\mathbf{T}^N(\mathbf{R}_{80}-\mathbf{R}_{20})J_0$ and $(\mathbf{R}_{80}-\mathbf{R}_{20})J_0$, and the elastic transmission \mathbf{T}^N can be easily determined from the ratio of these two difference spectra, similar to typical substrate variation

measurements performed with designed substrates. Supplementary Fig. 14c shows the accumulated proportion of these two distributions to clearly illustrate the essence of this virtual substrate measurement. Any pixel-sized “fictitious grains” that have the same accumulated proportion value in the range of 15%–85% can be considered as useful pixel-sized “fictitious grains”. The matrices representing electron–solid interactions of these pixel-sized “fictitious grains” at a given accumulated proportion value can be written as \mathbf{R}_x , which represents mapping from the incident spectrum to a new reflected spectrum of a pixel-sized “substrate” whose accumulated proportion value is x ($x = 0$ – 100). Therefore, any combination of two pixel-sized “fictitious grains” can be used to perform virtual substrate measurements, which means there should be infinite number of paired difference spectra $\mathbf{T}^N(\mathbf{R}_{x1}-\mathbf{R}_{x2})J_0$ and $(\mathbf{R}_{x1}-\mathbf{R}_{x2})J_0$, where $x1$ and $x2$ can be any number from 0 to 100 according to the levels of accumulated proportion. According to the definition of the virtual substrate method, measurements using different substrates should produce the same results, which means two accumulated proportion values in the bare gold substrate and graphene-covered regions should be linear except for those at levels that are too low ($< 15\%$) or too high ($> 85\%$). Very obvious linear relationships of these accumulated proportion values can be found for both the bare gold substrate and graphene-covered regions and the ratio of the two slopes of the lines of best fit represents \mathbf{T}^N of graphene. It should be noted that this \mathbf{T}^N determined from the ratio of two slopes can be considered as the average value from every possible combinations of $x1$ and $x2$ except those at levels that are too low ($< 15\%$) or too high ($> 85\%$). Based on this method of SEM image analysis, the determined \mathbf{T}^N at 10, 100, 200 and 400 eV were 0.68, 0.74, 0.85 and 0.90, respectively. These values are a little higher than those measured using a clean graphene/gold sample because of the carbon contamination on the present sample. Generally, the carbon contamination from residual organic glue is greater on gold surfaces than graphene because of the strong adsorption ability of gold surfaces, which increases the measured transmission value. This effect cannot be removed in a virtual substrate measurement. However, this example does reveal that any substrate that shows a relatively stable intensity distribution in a SEM image can be used to perform virtual substrate measurements, not only specially prepared polycrystalline or patterned substrates. In fact, almost every common substrate-supported nanomaterial except those on top of single crystals show a relatively stable intensity distribution in SEM images. Therefore, they can all be directly employed to perform virtual substrate measurements in which pure nanomaterial information at a given energy can be extracted without any additional procedures other than obtaining a SEM image.

Comment 2: *In the description of the method, present tense throughout would be most appropriate (in my opinion). Therefore: L127: "measurements were" should be "measurements are"; L143: "performed" should be "perform".*

Response to Comment 2: Thanks for your suggestions. In the revised manuscript, we used present tense to describe our method. Accordingly, "measurements were" and "performed" have been replaced by "measurements are" and "perform", respectively.

Comment 3: L191-196: *The "intuitive description" is valid only for thin layers, where it is appropriate to neglect the inelastic contribution. Perhaps this caveat should be included.*

Response to Comment 3: We apologize for omitting this point in our previous manuscript. We have added this sentence on page 9 as "It should be noted that this "intuitive description" is valid only for thin layers, where it is appropriate to neglect the inelastic contribution."

Comment 4: L208-220: *This section is not very clear. The terminology "approximate separation of energy channels" is unclear to a broad audience (me), as is "trackable white electrons" (the first phrase is clarified in the supplementary material). I've understood the white electron concept, but not from these lines.*

Response to Comment 4: We apologize for unclear terms. We have rewritten this paragraph on page 10.

- "In this method, the nanomaterial is back-illuminated by the reflected SE spectrum as a "white electron" probe analogous to the widely used white X-rays [15]. But unlike white X-rays, when white electrons are used as a probe, the information carried by the white electrons in the SE energy range is obscured by the undesired electron signals produced by energy loss or the formation of new SEs in inelastic collisions, while white electrons of higher energy travel in a target nanomaterial. Therefore, the energy channels in the measurement are approximately rather than fully separated when the reflected SEs from the underlying substrate are used as a white electron probe. In this case, the underlying substrate acts as a backscatterer of the monochromatic electron beam with fully dispersed energies, forming the white electrons. The transmission of white electrons through the nanomaterial enables the e–e interactions of the nanomaterial supported by the substrate to be studied with ultimate efficiency;..."

We also gave up using term "trackable white electron", instead we use more specific descriptions. For instance, "...however, white electrons are not typically tracked well in reflection-configuration measurements." on page 10 have been replaced by "however, the initial energy distributions of these white electrons are difficult to measure in reflection-configuration because of the influence of the reflection from the nanomaterial."

"...trackable white electrons can be created and controlled to quantitatively investigate..." on page 11 have been replaced by "..., white electrons originating from substrate-reflected electrons, can be controlled to quantitatively investigate ..."

Besides that, we have rewritten one sentence in the abstract in which "trackable" was used.

- “Implementing this method in secondary electron (SE) microscopy, a SE spectrum (white electrons) associated with the reflectivity difference between two different substrates can be **tracked** and controlled. The SE spectrum is used to quantitatively investigate the covering nanomaterial based on subtle changes in the transmission of the nanomaterial with extremely high efficiency rivaling that of conventional core-level electrons.”

Comment 5: L290: *"showed" should be "show."* Also, the sentence (L288-291) still makes it sound as though the authors have performed the point source microscopy, at least until one looks at the references. A rearrangement might be clearer: *"The virtual substrate measurements show excellent agreement with existing electron point source microscopy at 66 eV [23] and 100 eV [24] and with elastic transmission measurements performed in this work using the AES overlayer method [21] at 2.3 eV (gold surface plasmon), 78 eV (Si LVV), and 503 eV (O KLL) [22]."*

Response to Comment 5: We apologize for the unclear description in the original manuscript. We have rewritten this paragraph according to your suggestion on page 14 as follows:

- “**The virtual substrate measurements show excellent agreement with existing electron point source microscopy at 66 eV [22] and 100 eV [23] and with elastic transmission measurements performed in this work using the AES overlayer method [24] at 2.3 eV (gold surface plasmon) [25], 78 eV (Si LVV), and 503 eV (O KLL)**”.

Comment 6: L328 (and surrounding discussion): *Does "useless" signify the normal background signal? Are the "useful" data points considered to be those under a peak of limited extent in energy? This section could be written more clearly.*

Response to Comment 6: We apologize for the unclear description in the original manuscript. In order to avoid misleading, we have deleted this section and rewritten this paragraph on page 15 and 16 as follows:

- “Although multi-point probe measurements have been used to obtain information from measured spectra for many years already, like multi-spectral approaches using Auger electron microscopy [31], **the concept behind the virtual substrate measurement presented in this work is a new development.**”

Comment 7: L332: *"trackable white electrons" is still a mysterious phrase to me.*

Response to Comment 7: We apologize for the unclear term “trackable white electrons” in the original manuscript. We gave up using the “trackable” to describe the white electron, and use more specific description instead as listed in Response to Comment 4.

Comment 8: L500-501: I would suggest placing reference [23,24] after the word "microscopy," and inserting a reference to supplementary material after "method" (unless the LEEPS authors were the ones who used the MC method).

Response to Comment 8: We apologize for the unclear citations. We have placed the reference "[22,23]" ("[23,24]" in previous manuscript) after the word "microscopy" and one reference has been added in the "Transmission calculation" part in "Methods" section on page 33: "In the MC calculation [35], the elastic scattering was determined by the Mott cross section based on the muffin-tin model potential,...". The corresponding reference is added after the "Methods" section as follows:

[35] Ding, Z. J. & Shimizu, R. A Monte Carlo modeling of electron interaction with solids including cascade secondary electron production. *Scanning* **18**, 92 (1996).

Comment 9: *Supplementary Material:* L66: keV should be eV; L71: CAM should be CMA; Fig. 7: Flow chart top box should read "virtual crystal measurement".

Response to Comment 9: We apologize for our mistakes in supplementary material. We have replaced "keV" and "CMA" by "eV" and "CMA", respectively. We also have redrawn Supplementary Fig. 7, in which "Virtual substrate measurement" was used instead of "nano-chop-nod measurement" in the flow chart.